# An Improved Total and Tropospheric NO$_2$ Column Retrieval for GOME-2

Song Liu[1], Pieter Valks[1], Gaia Pinardi[2], Isabelle De Smedt[2], Huan Yu[2], Steffen Beirle[3], and Andreas Richter[4]

[1]Deutsches Zentrum für Luft- und Raumfahrt (DLR), Institut für Methodik der Fernerkundung (IMF), Oberpfaffenhofen, Germany
[2]Belgian Institute for Space Aeronomy (BIRA-IASB), Brussels, Belgium
[3]Max Planck Institute for Chemistry, Mainz, Germany
[4]Institute of Environmental Physics (IUP-UB), University of Bremen, Bremen, Germany

*Correspondence to:* Song Liu (Song.Liu@dlr.de)

**Abstract.** An improved algorithm for the retrieval of total and tropospheric nitrogen dioxide (NO$_2$) columns from the Global Ozone Monitoring Experiment-2 (GOME-2) is presented. The refined retrieval will be implemented in a future version of the GOME Data Processor (GDP) as used by the EUMETSAT Satellite Application Facility on Atmospheric Composition and UV Radiation (AC-SAF). The first main improvement is the application of an extended 425-497 nm wavelength fitting window in the differential optical absorption spectroscopy (DOAS) retrieval of the NO$_2$ slant column density, based on which initial total NO$_2$ columns are computed using stratospheric air mass factors (AMFs). Updated absorption cross-sections and a linear offset correction are used for the large fitting window. An improved slit function treatment is applied to compensate for both long-term and in-orbit drift of the GOME-2 slit function. Compared to the current operational (GDP 4.8) dataset, the use of these new features increases the NO$_2$ columns by $\sim$1-3 $\times 10^{14}$ molec/cm$^2$ and reduces the slant column error by $\sim$24%. In addition, the bias between GOME-2A and GOME-2B measurements is largely reduced by adopting a new level 1b data version in the DOAS retrieval. The retrieved NO$_2$ slant columns show good consistency with the Quality Assurance for Essential Climate Variables (QA4ECV) retrieval with a good overall quality. Second, the STRatospheric Estimation Algorithm from Mainz (STREAM), which was originally developed for the TROPOspheric Monitoring Instrument (TROPOMI) instrument, was optimized for GOME-2 measurements to determine the stratospheric NO$_2$ column density. Applied to synthetic GOME-2 data, the estimated stratospheric NO$_2$ columns from STREAM shows a good agreement with the a priori truth. An improved latitudinal correction is introduced in STREAM to reduce the biases over the subtropics. Applied to GOME-2 measurements, STREAM largely reduces the overestimation of stratospheric NO$_2$ columns over polluted regions in the GDP 4.8 dataset. Third, the calculation of AMF applies an updated box-air mass factor (box-AMF) look-up table (LUT) calculated using the latest version 2.7 of Vector-LInearized Discrete Ordinate Radiative Transfer (VLIDORT) model with an increased number of reference points and vertical layers, a new GOME-2 surface albedo climatology, and improved a priori NO$_2$ profiles obtained from the TM5-MP chemistry transport model. A large effect (mainly enhancement in summer and reduction in winter) on the retrieved tropospheric NO$_2$ columns by more than 10% is found over polluted regions. To evaluate the GOME-2 tropospheric NO$_2$ columns, an end-to-end validation is performed using ground-based multiple-axis DOAS (MAXDOAS) measurements.

The validation is illustrated for 6 stations covering urban, suburban, and background situations. Compared to the GDP 4.8 product, the new dataset presents an improved agreement with the MAXDOAS measurements for all the stations.

## 1 Introduction

Nitrogen dioxide ($NO_2$) is an important trace gas in the Earth's atmosphere. In the stratosphere, $NO_2$ is strongly related to halogen compound reactions and ozone destruction (Solomon, 1999). In the troposphere, nitrogen oxides ($NO_x$=$NO_2$+$NO$) serve as a precursor of zone in the presence of volatile organic compounds (VOC) and of secondary aerosol through gas-to-particle conversion (Seinfeld et al., 1998). As a prominent air pollutant affecting human health and ecosystem, large amounts of $NO_2$ are produced in the boundary layer by industrial processes, power generation, transportation, and biomass burning over polluted hot spots. For instance, a strong growth of $NO_2$ during the past two decades has caused severe air pollution problems for China with largest $NO_2$ columns in 2011, since then, cleaner techniques and stricter controlling have been applied to reduce the $NO_2$ pollution (Richter et al., 2005; Mijling et al., 2017; Liu et al., 2017). An increase in $NO_2$ concentrations due to economic growth is also found over India with a peak in 2012 (Hilboll et al., 2017). Despite the decrease in $NO_x$ emissions in Europe, still around half of European Union member states exceed the air quality standards mainly caused by diesel car emissions (European Commission, 2017).

$NO_2$ column measurements have been provided by satellite instruments, e.g., Global Ozone Monitoring Experiment (GOME) (Burrows et al., 1999), SCanning Imaging Absorption SpectroMeter for Atmospheric CHartographY (SCIAMACHY) (Bovensmann et al., 1999), Ozone Monitoring Instrument (OMI) (Levelt et al., 2006), and Global Ozone Monitoring Experiment-2 (GOME-2) (Callies et al., 2000; Munro et al., 2016). $NO_2$ observations will be continued by the new generation instruments with high spatial resolution such as TROPOspheric Monitoring Instrument (TROPOMI) (launched in October 2017, Veefkind et al., 2012) and by geostationary missions such as Sentinel-4 (Ingmann et al., 2012). The GOME-2 instrument, which is the main focus of this study, is included on a series of MetOp satellites as part of the EUMETSAT Polar System (EPS). The first GOME-2 was launched in October 2006 aboard the MetOp-A satellite, and a second GOME-2 was launched in September 2012 aboard MetOp-B. The consistent long-term dataset will be further extended by the third GOME-2 on the upcoming MetOp-C platform (to be launched in September 2018). $NO_2$ measurements from GOME-2 have been widely used to characterise the distribution, evolution, or transport of $NO_2$ (e.g., Hilboll et al., 2013, 2017; Zien et al., 2014), to estimate the $NO_x$ emission (e.g., Gu et al., 2014; Miyazaki et al., 2017; Ding et al., 2017), and to interpret VOC levels, ozone variation, or anthropogenic aerosol loading (e.g., Vrekoussis et al., 2010; Safieddine et al., 2013; Penning de Vries et al., 2015).

The GOME-2 total and tropospheric $NO_2$ products are generated using the GOME Data Processor (GDP) algorithm at the German Aerospace Center (DLR). The retrieval algorithm has been first described by Valks et al. (2011) as implemented in the GDP version 4.4 and was later updated to the current operational version 4.8 (Valks et al., 2017). The $NO_2$ retrieval for GOME-2 follows a classical 3-steps scheme.

First, the total $NO_2$ slant columns (namely the concentration integrated along the effective light path from the Sun through the atmosphere to the instrument) are derived using the differential optical absorption spectroscopy (DOAS) method (Platt and

Stutz, 2008). The DOAS technique is a least-squares method fitting the molecular absorption cross-sections to the measured GOME-2 sun-normalized radiances provided by the EUMETSAT's processing facility. The fit is applied on the data within a fitting window optimized for $NO_2$. As analysed by Richter et al. (2011) and in the Quality Assurance for Essential Climate Variables (QA4ECV, www.qa4ecv.eu) project, extension of the fitting window for GOME-2 increases the signal-to-noise ratio

and hence improves the $NO_2$ slant column error. The total $NO_2$ slant columns depend on the viewing geometry and also on parameters such as surface albedo and the presence of clouds and aerosol loads. They are therefore converted to initial total $NO_2$ vertical columns through division by a stratospheric airmass factor.

Second, the stratospheric contribution is estimated and separated from the $NO_2$ slant columns (referred to as stratosphere-troposphere separation). The GDP 4.8 algorithm applies a modified reference sector method, which uses measurements over

clean regions to estimate the stratospheric $NO_2$ columns based on the assumption of longitudinally invariable stratospheric $NO_2$ layers and of negligible tropospheric $NO_2$ abundance over the clean areas. The modified reference sector method defines a global pollution mask to remove potentially polluted regions and applies an interpolation over the unmask areas to derive the stratospheric $NO_2$ columns. As a result of using a fixed pollution mask, the modified reference sector method in GDP 4.8 has larger uncertainties over polluted areas, because limited amount of information over continents is used. To overcome the short-

comings, the STRatospheric Estimation Algorithm from Mainz (STREAM) method (Beirle et al., 2016) has been developed for TROPOMI instrument and was also successfully applied on GOME, SCIAMACHY, OMI, and GOME-2 measurements. Belonging also to the modified reference sector method, STREAM defines not a fixed pollution mask but weighting factors for each observation to determine its contribution to the stratospheric estimation.

Third, the tropospheric $NO_2$ vertical columns are calculated from the tropospheric slant columns by an air mass factor (AMF)

calculation, which contributes the largest uncertainty to the $NO_2$ retrieval, in particular over polluted regions (Boersma et al., 2004). The AMFs are determined with a radiative transfer model (RTM) and stored in a look-up table (LUT) requiring ancillary information such as surface albedo, vertical shape of the a priori $NO_2$ profile, clouds and aerosols. Improvements in the RTM and LUT interpolation scheme, the ancillary parameters, and the cloud and aerosol correction approach have been reported for OMI instrument (e.g., Boersma et al., 2011; Lorente et al., 2017; Vasilkov et al., 2017; Krotkov et al., 2017; Veefkind et al.,

2016; Lin et al., 2014; Castellanos et al., 2015; Laughner et al., 2018), which in principle are beneficial for similar satellite instruments like GOME-2.

In this paper, a new algorithm to retrieve the total and tropospheric $NO_2$ for the GOME-2 instruments is described, which includes improvements in each of the 3 algorithm steps introduced above. The improved algorithm will be implemented in the next version of GDP (referred to as GDP 4.9 hereafter). We briefly introduce the GOME-2 instrument (Sect. 2) and the current

operational (GDP 4.8) total and tropospheric $NO_2$ retrieval algorithm (Sect. 3). We present the improvements to the DOAS slant column retrieval (Sect. 4), the stratosphere-troposphere separation (Sect. 5), and the AMF calculation (Sect. 6). Finally, we show an end-to-end validation of the tropospheric $NO_2$ dataset using ground-based multiple-axis DOAS (MAXDOAS) datasets with different pollution conditions (Sect. 7).

## 2 Instrument and measurements

GOME-2 is a nadir-scanning UV-VIS spectrometer aboard the MetOp-A and MetOp-B satellites (referred to as GOME-2A and GOME-2B throughout this study) with a satellite repeating cycle of 29 days and an equator crossing time of 9:30 local time (descending node). The GOME-2 instrument measures the Earth's backscattered radiance and extra-terrestrial solar irradiance in the spectral range between 240 and 790 nm. The morning measurements from GOME-2 provide a better understanding of the diurnal variations of the $NO_2$ columns in combination with afternoon observations from for example the OMI and TROPOMI instruments (13:30 local time). The default swath width of GOME-2 is 1920 km, enabling a global coverage in $\sim$1.5 days. The default ground pixel size is $80\times40$ km$^2$ in the forward scan, which remains almost constant over the full swath width. In a tandem operation of MetOp-A and MetOp-B from July 2013 onwards, a decreased swath of 960 km and an increased spatial resolution of $40\times40$ km$^2$ are employed by GOME-2A. See Munro et al. (2016) for more details on instrument design and performance.

The operational GOME-2 $NO_2$ product is provided by DLR in the framework of EUMETSAT's Satellite Application Facility on Atmospheric Composition Monitoring (AC-SAF). The product processing chain starts with the level 0 to 1b processing within the core ground segment at EUMETSAT in Darmstadt (Germany), where the raw instrument (level 0) data is converted into geolocated and calibrated (level 1b) (ir)radiances by the GOME-2 Product Processing Facility (PPF). The level 1b (ir)radiances are disseminated through the EUMETCast system to the AC-SAF processing facility at DLR in Oberpfaffenhofen (Germany), and further processed using the Universal Processor for UV/VIS Atmospheric Spectrometers (UPAS) system. Broadcasted via EUMETCast, WMO/GTS, and the Internet, the resulting level 2 near-real-time total column products including $NO_2$ columns can be received by user communities 2 hours after sensing. Offline and reprocessed GOME-2 level 2 and consolidated products are also provided within 1 day by DLR, which can be ordered via FTP-server and the EUMETSAT Data Centre (https://acsaf.org/).

## 3 Total and tropospheric NO$_2$ retrieval for GDP 4.8

The first main step of the retrieval algorithm is the DOAS technique, which is applied to determine the total $NO_2$ slant columns from the (ir)radiance spectra measured by the instrument. Based on the Beer-Lambert's law, the DOAS fit is a least-squares inversion to isolate the trace gas absorption from the background processes, e.g., extinction resulting from scattering on molecules and aerosols, with a background polynomial $P(\lambda)$ at wavelength $\lambda$:

$$\ln\left[\frac{I(\lambda)+offset(\lambda)}{I^0(\lambda)}\right] = -\sum_g S_g\sigma_g(\lambda) - \alpha_R R(\lambda) - P(\lambda). \tag{1}$$

The measurement-based term is defined as the natural logarithm of the measured earthshine radiance spectrum $I(\lambda)$ divided by the daily solar irradiance spectrum $I^0(\lambda)$. The intensity offset correction $offset(\lambda)$, which describes the additional contributions such as stray light in the spectrometer to the measured intensity, is modelled using a zero order polynomial with polynomial coefficient as fitting parameter. The spectral effect from the absorption of species $g$ is determined by the fitted slant

**Table 1.** Main settings of GOME-2 DOAS retrieval of $NO_2$ slant columns discussed in this study.

|  | GDP 4.8 (Valks et al., 2011, 2017) | GDP 4.9 (this work) | QA4ECV (Müller et al., 2016; Boersma et al., 2018) |
|---|---|---|---|
| Wavelength range | 425-450 nm | 425-497 nm | 405-465 nm |
| Cross-sections | $NO_2$ 240K, $H_2O_{vap}$, $O_3$, $O_4$, Ring | $NO_2$ 220K, $H_2O_{vap}$, $O_3$, $O_4$, Ring, $H_2O_{liq}$, Eta, Zeta, resol correction | $NO_2$ 220K, $H_2O_{vap}$, $O_3$, $O_4$, Ring, $H_2O_{liq}$ |
| Polynomial degree | 3 | 5 | 5 |
| Intensity offset | Constant | Linear | Constant |
| Slit function | Preflight | Stretched preflight | Preflight |

column density $S_g$ and associated absorption cross-section $\sigma_g(\lambda)$. An additional term with the Ring scaling factor $\alpha_R$ and the Ring reference spectrum $R(\lambda)$ describes the filling-in effect of Fraunhofer lines by rotational Raman scattering (the so-called Ring effect). The GDP 4.8 algorithm adopts a wavelength range of 425-450 nm to ensure prominent $NO_2$ absorption structures and controllable interferences from other absorbing species, e.g., water vapor ($H_2O_{vap}$), ozone ($O_3$), and oxygen dimer ($O_4$). Table 1 gives an overview of the DOAS settings for the current operational GDP 4.8 algorithm, the improved version 4.9 algorithm (see Sect. 4), and the algorithm used in the QA4ECV product (see Sect. 4.5).

The second component in the retrieval is the calculation of initial total vertical column densities $V_{init}$ using an stratospheric AMF ($M_{strat}$) conversion:

$$V_{init} = \frac{S}{M_{strat}}. \tag{2}$$

Given the small optical thickness of $NO_2$, $M_{strat}$ can be determined as:

$$M_{strat} = \frac{\sum_l m_l(\boldsymbol{b}) x_l c_l}{\sum_l x_l} \tag{3}$$

with $m_l$ the box-air mass factors (box-AMFs) in layer $l$, $x_l$ the altitude-dependent subcolumns from a stratospheric a priori $NO_2$ profiles climatology (Lambert et al., 1999), and $c_l$ a correction coefficient to account for the temperature dependency of $NO_2$ cross-section (Boersma et al., 2004; Nüß et al., 2006). The calculation of $V_{init}$ assumes negligible tropospheric $NO_2$ and hence uses only the stratospheric a priori $NO_2$ profiles to derive AMF. The box-AMFs $m_l$ are derived using the multi-layered multiple scattering LIDORT RTM (Spurr et al., 2001) and stored in a LUT as a function of various model inputs $\boldsymbol{b}$, including GOME-2 viewing geometry, surface pressure, and surface albedo. The surface albedo is described by the Lambertian-equivalent reflectivity (LER). The surface LER climatology used in the GDP 4.8 algorithm is derived from combined TOMS/GOME measurements (Boersma et al., 2004) for the years 1979-1993 with a spatial resolution of $1.25°\text{lon} \times 1.0°\text{lat}$.

In the presence of clouds, the calculation of $M_{strat}$ adopts the independent pixel approximation based on GOME-2 cloud parameters:

$$M_{strat} = \omega M_{strat}^{cloud} + (1 - \omega) M_{strat}^{clear} \tag{4}$$

with $\omega$ the cloud radiance fraction, $M_{strat}^{cloud}$ the cloudy-sky stratospheric AMF, and $M_{strat}^{clear}$ the clear-sky stratospheric AMF. $M_{strat}^{cloud}$ and $M_{strat}^{clear}$ are derived with Eq. (3) with $M_{strat}^{cloud}$ mainly relying on the cloud pressure and the cloud albedo. $\omega$ is derived from the cloud fraction $c_f$:

$$\omega = \frac{c_f I^{cloud}}{(1 - c_f) I^{clear} + c_f I^{cloud}}, \tag{5}$$

where $I^{cloud}$ is the radiance for a cloudy scene and $I^{clear}$ for a clear scene. $I^{cloud}$ and $I^{clear}$ are calculated using LIDORT, depending mostly on the GOME-2 viewing geometry, surface albedo and cloud albedo. From GOME-2, $c_f$ is determined with the Optical Cloud Recognition Algorithm (OCRA) by separating a spectral scene into cloudy contribution and cloud-free background, and the cloud pressure and the cloud albedo are derived using the Retrieval Of Cloud Information using Neural Networks (ROCINN) algorithm by comparing simulated and measured radiance in and near the $O_2$ A-band (Loyola et al., 2007,

2011). Applied in the $NO_2$ retrieval in GDP 4.8, the latest version 3.0 of the OCRA (Lutz et al., 2016) applies a degradation correction on the GOME-2 level 1 measurements as well as corrections for viewing angle and latitudinal dependencies. A new cloud-free background is constructed from six years of GOME-2A measurements from the years 2008-2013. The updated OCRA also includes an improved detection and removal of sun glint that affects most of the GOME-2 orbits. The version 3.0 of ROCINN (Loyola et al., 2018) applies a forward RTM calculation using updated surface albedo climatology and spectroscopic

data as well as a new inversion scheme based on Tikhonov regularization (Tikhonov and Arsenin, 1977; Doicu et al., 2010). The computation time of ROCINN is optimised with a smart sampling method (Loyola et al., 2016).

    The next retrieval step is the separation of stratospheric and tropospheric components from the initial vertical total columns, namely the "stratosphere-troposphere separation". Since no direct stratospheric measurements are available for GOME-2, a spatial filtering algorithm is applied to estimate the stratospheric $NO_2$ columns in GDP 4.8. The spatial filtering algorithm belongs

to the modified reference sector method, which uses total $NO_2$ columns over clean regions to approximate the stratospheric $NO_2$ columns based on the assumption of longitudinally invariable stratospheric $NO_2$ layers and of negligible tropospheric $NO_2$ abundance over the clean areas. The spatial filtering algorithm uses a pollution mask to filter the potentially polluted areas (tropospheric $NO_2$ columns larger than $1 \times 10^{15}$ molec/cm$^2$), followed by a low-pass filtering (with a zonal $30°$ boxcar filter) on the initial total columns of the unmasked areas, and afterwards a removal of a tropospheric background $NO_2$ ($1 \times 10^{14}$

molec/cm$^2$) from the derived stratospheric columns.

    Finally, the tropospheric $NO_2$ columns $V_{trop}$ can be computed as:

$$V_{trop} = \frac{M_{strat}}{M_{trop}} \times T, \tag{6}$$

where $M_{strat}$ is the stratospheric AMF in Eq. (3), $M_{trop}$ is the tropospheric AMF, and $T$ is the tropospheric residues ($T = V_{init} - V_{strat}$). $M_{trop}$ is determined using Eq. (3) and (4) with tropospheric a priori $NO_2$ profiles. The calculation of

$M_{trop}$ relies on the same model parameters as of $M_{strat}$, but the dependency on the parameters like surface albedo and cloud properties as well as on the a priori $NO_2$ profiles is much stronger. The GDP 4.8 adopts the tropospheric a priori $NO_2$ profiles from a run of global chemistry transport model MOZART version 2 (Horowitz et al., 2003) with anthropogenic emissions from the EDGAR2.0 inventory (Olivier et al., 1996) for the early 1990s. The monthly average vertical profiles are calcu-

lated from MOZART-2 data from the year 1997 for the overpass time of GOME-2 (9:30 local time) with a resolution of $1.875°\text{lon} \times 1.875°\text{lat}$.

## 4 Improved DOAS slant column retrieval

A larger 425-497 nm wavelength fitting window for the DOAS method (Richter et al., 2011) is implemented in the GDP 4.9 to retrieve the $NO_2$ slant columns, which improves the signal-to-noise ratio by including more $NO_2$ absorption structures. Compared to the extended 405-465 nm range, as employed by the QA4ECV GOME-2 $NO_2$ product and used in the $NO_2$ retrieval for OMI instrument (Boersma et al., 2002; van Geffen et al., 2015), the 425-497 nm fitting window has stronger sensitivity to $NO_2$ columns in boundary layer because the importance of scattering decreases with wavelength (Richter and verification team, 2015). In this study, the slant columns are derived using QDOAS software developed at the Belgian Institute for Space Aeronomy (BIRA-IASB) (Danckaert et al., 2015) [1]. Table 1 summarises the new settings of the GDP 4.9 algorithm.

### 4.1 Absorption cross-sections

In the fitting window optimized for $NO_2$ retrieval, the DOAS fit includes species with strong and unique absorption structures and describes their spectral effect using absorption cross-sections from literature. In our GDP 4.9 algorithm, the absorption cross-sections of $NO_2$, $H_2O_{vap}$, $O_3$, and $O_4$ are updated mainly with newly released datasets as:

- $NO_2$ absorption at 220K from Vandaele et al. (2002)

- $O_3$ absorption at 228K from Brion et al. (1998)

- $H_2O_{vap}$ absorption at 293K from HITEMP Rothman et al. (2010), rescaled as in Lampel et al. (2015)

- $O_4$ absorption at 293K from Thalman and Volkamer (2013)

In addition, to compensate for the larger spectral interference from liquid water ($H_2O_{liq}$), a $H_2O_{liq}$ absorption (Pope and Fry, 1997) is included to reduce systematic errors above ocean for the wider wavelength range. Two additional GOME-2 polarization key data (EUMETSAT, 2009) are included to correct for remaining polarization correction problems, particularly for GOME-2B.

- $H_2O_{liq}$ absorption at 297K from Pope and Fry (1997), smoothed as in Peters et al. (2014)

- Eta and Zeta from GOME-2 calibration key data (EUMETSAT, 2009)

It is worth noting that our improved DOAS retrieval in the GDP 4.9 adopts a decreased temperature of $NO_2$ cross-section (220K instead of 240K in GDP 4.8, Valks et al. 2017) for a consistency with other $NO_2$ retrievals from GOME-2, OMI and TROPOMI (Müller et al., 2016; Boersma et al., 2002; van Geffen et al., 2015, 2016), with minor effect on the fit quality

---

[1]Note that the derived slant columns are scaled by geometric AMFs to correct for the angular dependencies of GOME-2 measurements in this section.

($\sim 0.02\%$) from the two temperatures. Changing the temperature of $NO_2$ cross-section from 240K to 220K reduces the $NO_2$ slant columns by $\sim 6\%$-$9\%$, but this temperature dependency is corrected in the AMF and vertical column calculation (see Eq. (3)).

The spectral signature of sand absorption has been investigated by Richter et al. (2011) for GOME-2 data, but it is not applied here because of the potential interference with the broadband liquid water structure (Peters et al., 2014), which might lead to non-physical results over the ocean.

## 4.2 Intensity offset correction

Besides the radiances backscattered by the Earth's atmosphere, a number of both natural (i.e. the Ring effect) and instrumental (e.g., stray light in the spectrometer and change of detector's dark current) sources contribute to an additional "offset" to the scattering intensity. To correct for this drift, an intensity offset correction with a linear wavelength dependency (i.e., polynomial degree of 1) is applied for the large fitting window in this study. Figure 1 illustrates the effect of using a linear intensity offset correction for the large fitting window on 3 March 2008. The use of a linear offset correction increases the $NO_2$ columns by up to $3 \times 10^{14}$ molec/cm$^2$ (17%) and decreases the fitting residues (retrieval root-mean-square, RMS) by up to 30%. Larger differences are found at the eastern scans (eastern part of GOME-2 swath), possibly suggesting instrumental issues specific to GOME-2. For the retrieval RMS, stronger improvements are mainly located above ocean, arguably from the compensation of inelastic vibrational Raman scattering in water bodies (Vountas et al., 2003).

The intensity offset can also be fitted using only the constant term, as employed by the GDP 4.8 algorithm (with 425-450 nm wavelength window) and as recommended by the QA4ECV algorithm (with 405-465 nm). Compared to the use of linear intensity offset correction, the application of a constant term on our retrieval shows a decrease in the $NO_2$ columns by up to $3.5 \times 10^{14}$ molec/cm$^2$ (17%) and an increase in the retrieval RMS by up to 14%, which implies the necessity of using a linear intensity offset correction for the large 425-497 nm wavelength range.

## 4.3 GOME-2 slit function treatment

An accurate treatment of the instrumental slit function is essential for the wavelength calibration and the convolution of high-resolution laboratory cross-sections. In spite of a generally good spectral stability of GOME-2 in orbit, the width of GOME-2 slit function has been changing on both long and short timescales (Munro et al., 2016), which needs to be accounted for in the DOAS analysis. In this study, an improved treatment of GOME-2 slit function in the DOAS fit is achieved by calculating effective slit functions from GOME-2 irradiance measurements to correct for the long-term variations (see Sect. 4.3.1) and by including an additional cross-section in the DOAS fit to correct for the short-term variations (see Sect. 4.3.2).

### 4.3.1 Long-term variations

To analyse the long-term variations of the GOME-2 instrumental slit function and the impact on our retrieval, effective slit functions are derived by convolving a high-resolution reference solar spectrum (Chance and Kurucz, 2010) with a stretched

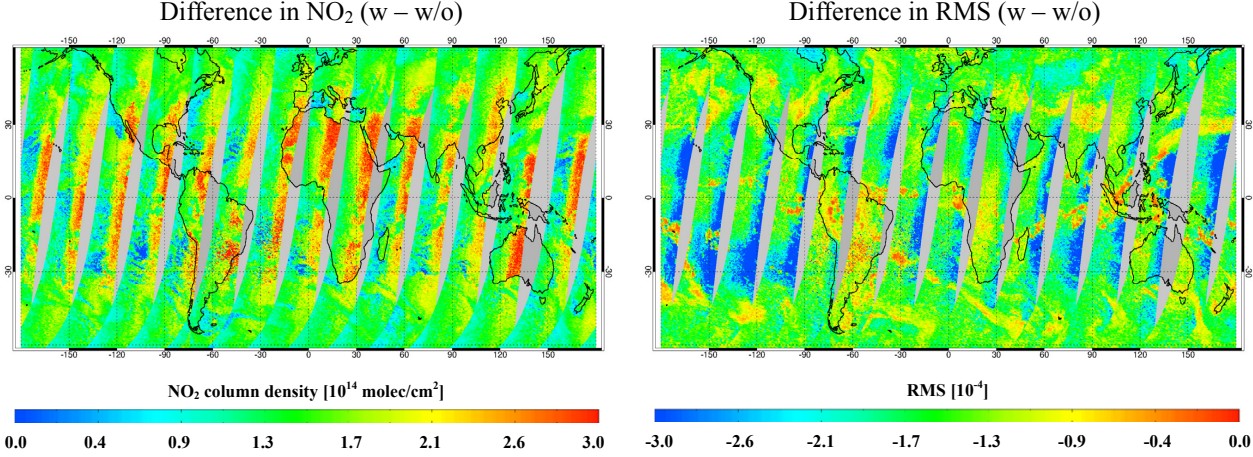

**Figure 1.** Difference in NO$_2$ columns (slant columns scaled by geometric AMFs) (left) and retrieval RMS (right) estimated with and without a linear intensity offset correction for GOME-2A on 3 March 2008.

preflight GOME-2 slit function and aligning to the GOME-2 daily irradiance measurements with stretch factors as fit parameters. The effective slit functions are calculated in 13 subwindows covering the full fitting window (425-497 nm). Figure 2 displays the long-term evolution of the fitted GOME-2 slit function width (full width at half maximum, FWHM) calculated from the stretch factors. The GOME-2 slit function has narrowed after the launch by ∼5% for GOME-2A and ∼3.5% for GOME-2B at 451 nm, in agreement with Dikty et al. (2011), Azam et al. (2015), and Munro et al. (2016). For GOME-2A, visible discontinuities of the slit function width are related to the in-orbit instrument operations, including an apparent anomaly in September 2009 when a major throughput test was performed (EUMETSAT, 2012). After the throughput test, the narrowing of slit function has slowed down. For GOME-2B, stronger seasonal fluctuations of the FWHM are found. The seasonal and long-term variations in the GOME-2 slit function are caused by changing temperatures of the optical bench due to the seasonal variation in solar heating and the lack of thermal stability of the optical bench, respectively (Munro et al., 2016). Although the variations are only a few percent, the effect on the DOAS retrieval is significant. Compared to the application of the preflight slit function, the use of a stretched slit function improves the calibration residuals by ∼40% for both GOME-2A and GOME-2B (not shown).

In previous studies, slit functions have also been fitted using various Gaussian shapes. For instance, De Smedt et al. (2012) have derived effective GOME-2 slit functions for formaldehyde retrieval using an asymmetric Gaussian with it's width and shape as fit parameters. For NO$_2$ retrieval, the use of effective slit functions with an asymmetric Gaussian leads to similar results as using a preflight slit function. In addition, Beirle et al. (2017) have proposed a slit function parameterization using a Super Gaussian, which is proved to quickly and robustly describe the slit function changes for satellite instrument OMI or TROPOMI. In the case of GOME-2, the Super Gaussian obtains nearly identical results as the asymmetric Gaussian and is therefore not applied in here.

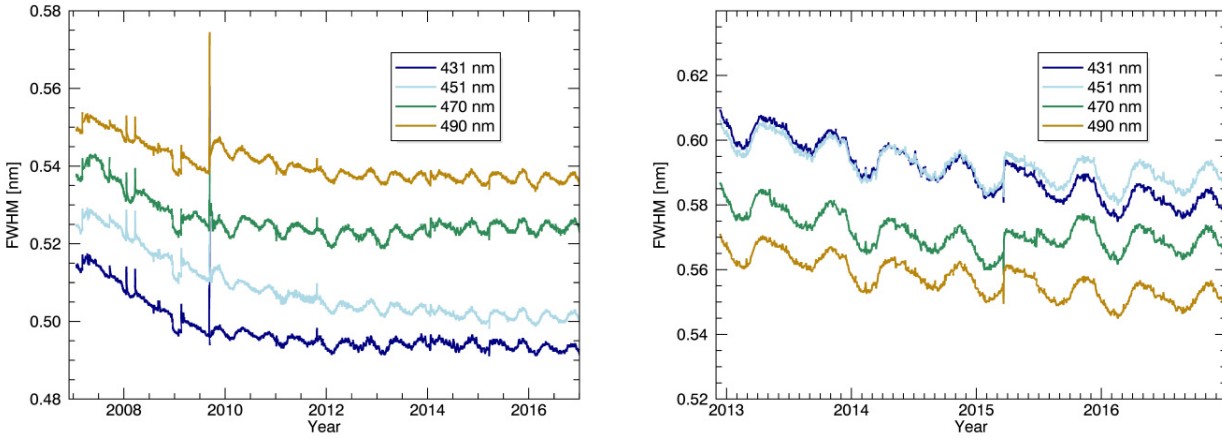

**Figure 2.** Temporal evolution of the fitted slit function FWHM for GOME-2A (left, January 2007-December 2016) and GOME-2B (right, December 2012-December 2016.)

### 4.3.2 In-orbit variations

To correct for the in-orbit variations of GOME-2 slit function, a "resolution correction function" (Azam et al., 2015) is included as an additional cross-section in the DOAS fit (see Table 1). The cross-section is derived by dividing a high-resolution solar spectrum (Chance and Kurucz, 2010) convolved with a stretched preflight GOME-2 slit function (see Sect. 4.3.1) by itself but
convolved with a slightly modified slit function. Figure 3 shows an example of the fit coefficients and the influence on our DOAS retrieval on 1 February 2013. As shown in the left panel, the slit function width increases along the orbit by $\sim 2 \times 10^{-3}$ nm ($\sim 0.4\%$) for GOME-2A (see Beirle et al. 2017, Fig. 8 therein) and $\sim 5.2 \times 10^{-3}$ nm ($\sim 1\%$) for GOME-2B (a fit coefficient of $1 \times 10^{-2}$ corresponds to a change in the slit function width of $\sim 2.8 \times 10^{-3}$ nm). This in-orbit broadening of the slit function is caused by the increasing temperature of the instrument along the orbit. Taking into account the in-orbit broadening in the
DOAS fit decreases the retrieval RMS by up to 5% for GOME-2A and up to 12% for GOME-2B in Fig. 3 (right).

### 4.4 GOME-2 level 1b data

As described in Sect. 2, the level 0 to 1b processing by the PPF at EUMETSAT calculates the geolocation and calibration parameters and produces the calibrated level 1b (ir)radiances. Due to the incomplete removal of Xe-line contamination in the GOME-2B calibration key-data (calibration key-data is taken during the on-ground campaign and required as an input to the
level 0 to 1b processing), artefacts at wavelength larger than 460 nm have been reported by Azam et al. (2015) for GOME-2B irradiances. Mainly focusing on the cleaning of contamination in the GOME-2B calibration key-data, a new version 6.1 of the GOME-2 level 0 to 1b processor has been activated from 25 June 2015 onwards (EUMETSAT, 2015). To study the impact of the new level 1b data on our GDP 4.9 algorithm using the 425-497 nm fitting window, the retrieval is analysed using both

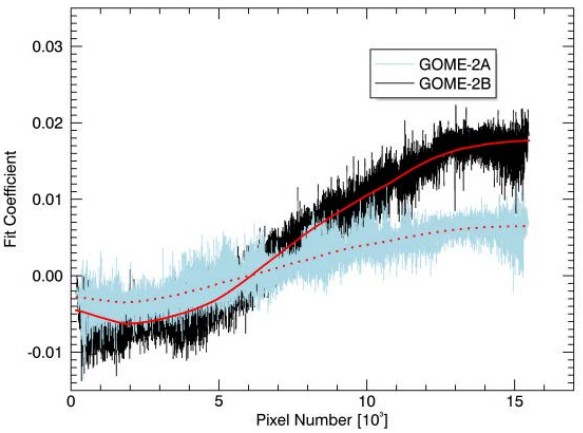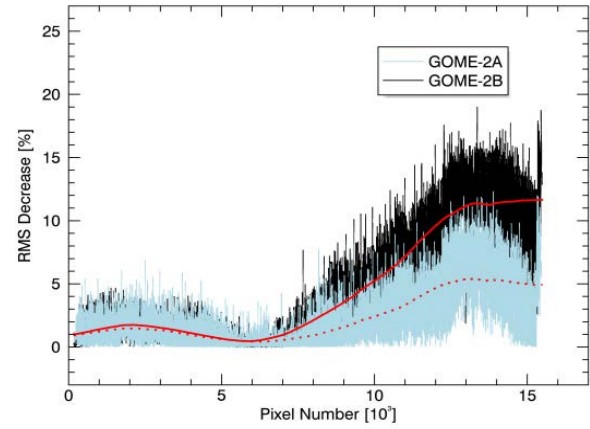

**Figure 3.** Changes of GOME-2 slit function width along orbit 32636 on 1 February 2013 (left) and the impact on the retrieval RMS error (right). Red lines provide the boxcar average for GOME-2A (dotted) and GOME-2B (solid). A fit coefficient of $1 \times 10^{-2}$ corresponds to a change in the slit function width of $\sim 2.8 \times 10^{-3}$ nm in the left panel.

the new version 6.1 (testing dataset provided by EUMETSAT for March 2015) and the previous version 6.0 data for the same period. Figure 4 presents a comparison of the retrieved $NO_2$ columns over the Pacific for GOME-2A and GOME-2B. The application of the version 6.1 level 1b data slightly reduces the $NO_2$ columns by $\sim 1\text{-}1.5 \times 10^{14}$ molec/cm$^2$ ($\sim 6\text{-}11\%$) for GOME-2A. A larger effect is observed for GOME-2B with a decrease of $NO_2$ columns by $\sim 3\text{-}4 \times 10^{14}$ molec/cm$^2$ ($\sim 15\text{-}23\%$)

and a reduction of RMS error by $\sim 27\text{-}33\%$ (not shown). The stronger decrease of GOME-2B $NO_2$ columns leads to a better consistency between the datasets from GOME-2A and GOME-2B with an overall bias reduced from $\sim 3 \times 10^{14}$ molec/cm$^2$ to $\sim 1 \times 10^{14}$ molec/cm$^2$.

### 4.5 Comparison to QA4ECV data

The quality of the GDP 4.9 retrieval is evaluated using the GOME-2 $NO_2$ dataset from QA4ECV, which is a project aiming at

10 quality-assured satellite products using a retrieval algorithm harmonised for GOME, SCIAMACHY, OMI and GOME-2. The GOME-2A $NO_2$ columns from QA4ECV (version 1.1) for the years 2007-2015 have shown an improved quality over previous datasets (Zara et al., 2018). Table 1 gives an overview of the DOAS settings used in the QA4ECV project. Figure 5 shows a comparison of the $NO_2$ columns over the Pacific from the GDP 4.8 algorithm, the GDP 4.9 algorithm, and the QA4ECV data for February 2007. For comparison, only ground pixels with solar zenith angle smaller than $80°$ are considered. The GDP

4.8 dataset has been adjusted using a 220K Vandaele et al. (2002) $NO_2$ cross-section to remove the influence of temperature dependency of $NO_2$ cross-section (see discussion in Sect. 4.1). Compared to the GDP 4.8 dataset, the improved DOAS retrieval in the GDP 4.9 increases the $NO_2$ columns by $\sim 1\text{-}3 \times 10^{14}$ molec/cm$^2$ (up to 27%). Compared to the QA4ECV product, a good overall consistency is found with the GDP 4.9 dataset at all latitudes considering the different DOAS settings such as fitting window, offset correction, and slit function characterisation.

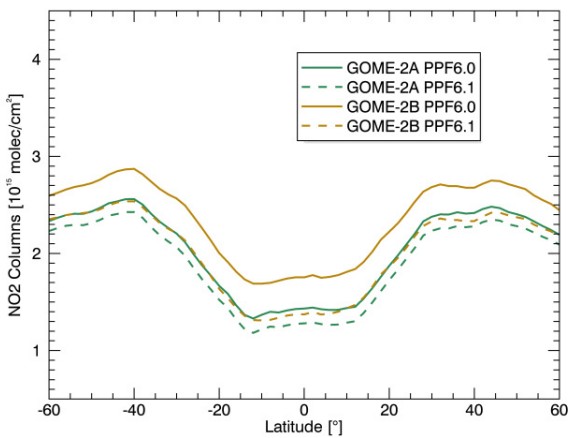

**Figure 4.** Monthly zonal average NO$_2$ columns (slant columns scaled by geometric AMFs) for GOME-2A (green) and GOME-2B (brown) using the new PPF 6.1 (dotted) and PPF 6.0 (solid) data in March 2015 over the Pacific (160°E-180°E).

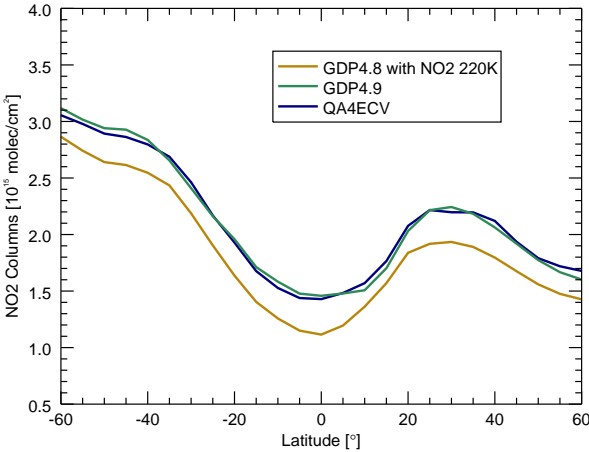

**Figure 5.** Comparisons of monthly zonal average NO$_2$ columns (slant columns scaled by geometric AMFs) from the operational GDP 4.8 product (but retrieved using a 220K NO$_2$ cross-section from Vandaele et al. 2002) (brown), the improved GDP 4.9 algorithm (green), and the QA4ECV dataset (blue) over the Pacific (160°E-180°E) in February 2007 for GOME-2A.

Figure 6 presents the time series of calculated slant column errors from the three datasets, following a statistical method to analyze the NO$_2$ slant column uncertainty for GOME-2 (Valks et al., 2011, Sect. 6.1 therein). The slant column errors, calculated as variations of NO$_2$ measurements within small boxes (2°×2°) over the tropical Pacific (20°S-20°N, 160°E-180°E), increase for all the three datasets as a result of instrument degradation (Dikty et al., 2011; Munro et al., 2016) until the major throughput test in September 2009 (see Sect. 4.3.1) and stabilize afterwards. Mainly driven by the use of a wider fitting window with stronger absorptions, smallest slant column errors are found by the GDP 4.9 algorithm, e.g., 23.8% smaller than from the

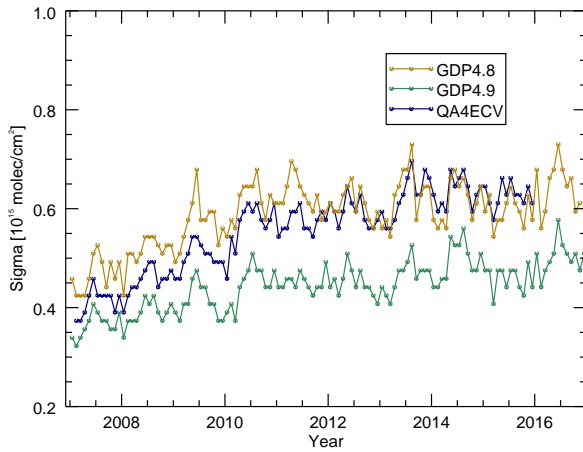

**Figure 6.** Temporal evolution of the $NO_2$ slant columns errors from the operational GDP 4.8 product (brown, January 2007-December 2016), the improved GDP 4.9 algorithm (green, January 2007-December 2016), and the QA4ECV dataset (blue, February 2007-December 2015) for GOME-2A, using deviations of $NO_2$ slant columns from box ($2° \times 2°$) mean values over the tropical Pacific (20°S-20°N, 160°E-180°E).

GDP 4.8 and 13.5% smaller than from the QA4ECV dataset in February 2007, with an increasing difference with time for the QA4ECV dataset (27.9% in December 2015).

## 5  New stratosphere-troposphere separation

The calculation of tropospheric $NO_2$ requires an estimation and removal of the stratospheric contribution to the initial total
$NO_2$ columns. In our GDP 4.9 retrieval, the stratosphere-troposphere separation algorithm STREAM (Beirle et al., 2016) has been adapted to GOME-2 measurements. Belonging to the modified reference sector method, STREAM uses initial total $NO_2$ columns with negligible tropospheric contribution, i.e., unpolluted measurements at remote areas and cloudy measurements at medium altitudes, to derive the stratospheric $NO_2$ columns. Based on a tropospheric $NO_2$ climatology and the GOME-2 cloud product, STREAM calculates weighting factors for each satellite pixel to define the contribution of initial total columns to the
stratospheric estimation: potentially polluted pixels are weighted low instead of being totally masked out in the GDP 4.8 spatial filtering method; cloudy observations at medium altitudes are given higher weights because they directly provide the stratospheric information; the weights are further adjusted in a second iteration if pixels suffer from large biases in the tropospheric residues. Depending on these weighting factors, stratospheric $NO_2$ fields are derived by weighted convolution on the daily initial total columns using convolution kernels. The convolution kernels are wider at lower latitudes due to the longitudinal
homogeneity assumption of stratospheric $NO_2$ and narrower at higher latitudes to reflect the stronger natural variations. To remove the biases in the weighted convolution resulting from the large latitudinal gradients, a latitudinal correction is applied on the initial total columns: the latitudinal dependencies of initial total $NO_2$ are calculated over the clean Pacific, removed from the initial total $NO_2$ before weighted convolution, and added back to the estimated stratospheric columns afterwards. However,

we found that longitudinal variations of $NO_2$ concentration resulted in biases in the latitudinal correction and hence in the stratospheric estimation. For the adaptation of STREAM to GOME-2 measurements, the performance of STREAM is analysed using synthetic GOME-2 $NO_2$ observations (see Sect. 5.1) and an improved latitudinal correction is applied (see Sect. 5.2).

## 5.1 Performance of STREAM

To test the performance of STREAM for GOME-2, simulated $NO_2$ fields from the C-IFS-CB05-BASCOE (referred to as C-IFS throughout this work) experiment (Huijnen et al., 2016) are applied. The C-IFS model is a combination of tropospheric chemistry module in the Integrated Forecast System (IFS, with current version based on the Carbon Bond chemistry scheme, CB05) of the European Centre for Medium-Range Weather Forecasts (ECMWF) and stratospheric chemistry from the Belgian Assimilation System for Chemical ObsErvations (BASCOE) system. Based on one year of C-IFS data (2009) at a resolution

of $0.75°\mathrm{lon}\times0.75°\mathrm{lat}$, synthetic initial total columns $V_{init}$ are calculated as:

$$V_{init} = \frac{S}{M_{strat}} = \frac{V_{total} \times M_{total}}{M_{strat}} \tag{7}$$

(see Eq. (2)). Modelled $NO_2$ slant columns $S$ are based on the total vertical columns $V_{total}$ from C-IFS with interpolation to match the GOME-2 centre pixel coordinate and measurement time. Total AMFs $M_{total}$ and stratospheric AMFs $M_{strat}$ are derived using Eq. (3)-(5) with surface properties and cloud information from GOME-2 orbital data and with C-IFS a priori

$NO_2$ profiles for the whole atmosphere and between the tropopause (defined by a latitude-dependent parameterization with the tropopause height ranging from 270 hPa for arctic to 92 hPa for tropics) and the top of the atmosphere, respectively. The performance of STREAM is evaluated by applying the synthetic initial total $NO_2$ columns and comparing the estimated stratospheric $NO_2$ columns with the a priori truth (stratospheric fields from C-IFS integrated between the tropopause and the top of the atmosphere).

Figure 7 displays the synthetic initial total columns from C-IFS, the modelled stratospheric columns, and the estimated stratospheric columns from STREAM on 5 February and 5 August 2009. The result from STREAM presents an overall smooth stratospheric pattern with a strong latitudinal and seasonal dependency resulting from photochemical changes and dynamical variabilities. Because the stratospheric values over polluted regions are taken from the clean measurements at the same latitude, the stratospheric and tropospheric contribution over polluted regions is well separated by STREAM, especially in the northern

hemisphere. Due to the latitude-dependent definition of convolution kernels, STREAM conserves the longitudinal gradients of stratospheric $NO_2$ at low latitudes and identifies certain strong stratospheric variations at high latitudes, e.g., in the polar vortex on 5 February. However, smaller structures in the synthetic initial total columns, for instance, resulting from the diurnal variation of $NO_2$ across an orbital swath, are aliased into the troposphere by STREAM due to the use of convolution kernels.

    Figure 8 (top) shows the differences in estimated (Fig. 7 bottom) and a priori (Fig. 7 center) stratospheric $NO_2$. Overall, the

stratospheric columns estimated from STREAM show a good agreement with the modelled truth with a slight overestimation, e.g., by $\sim$1-2$\times10^{14}$ molec/cm$^2$ over low latitudes for both days. Larger differences are found at higher latitudes, especially in winter, e.g., by $\sim5\times10^{14}$ molec/cm$^2$ over eastern Europe and over the North Pacific (west of Canada) on 5 February. The strong longitudinal variations of $NO_2$ over these regions in the a priori truth (Fig. 7 center) can not be completely captured

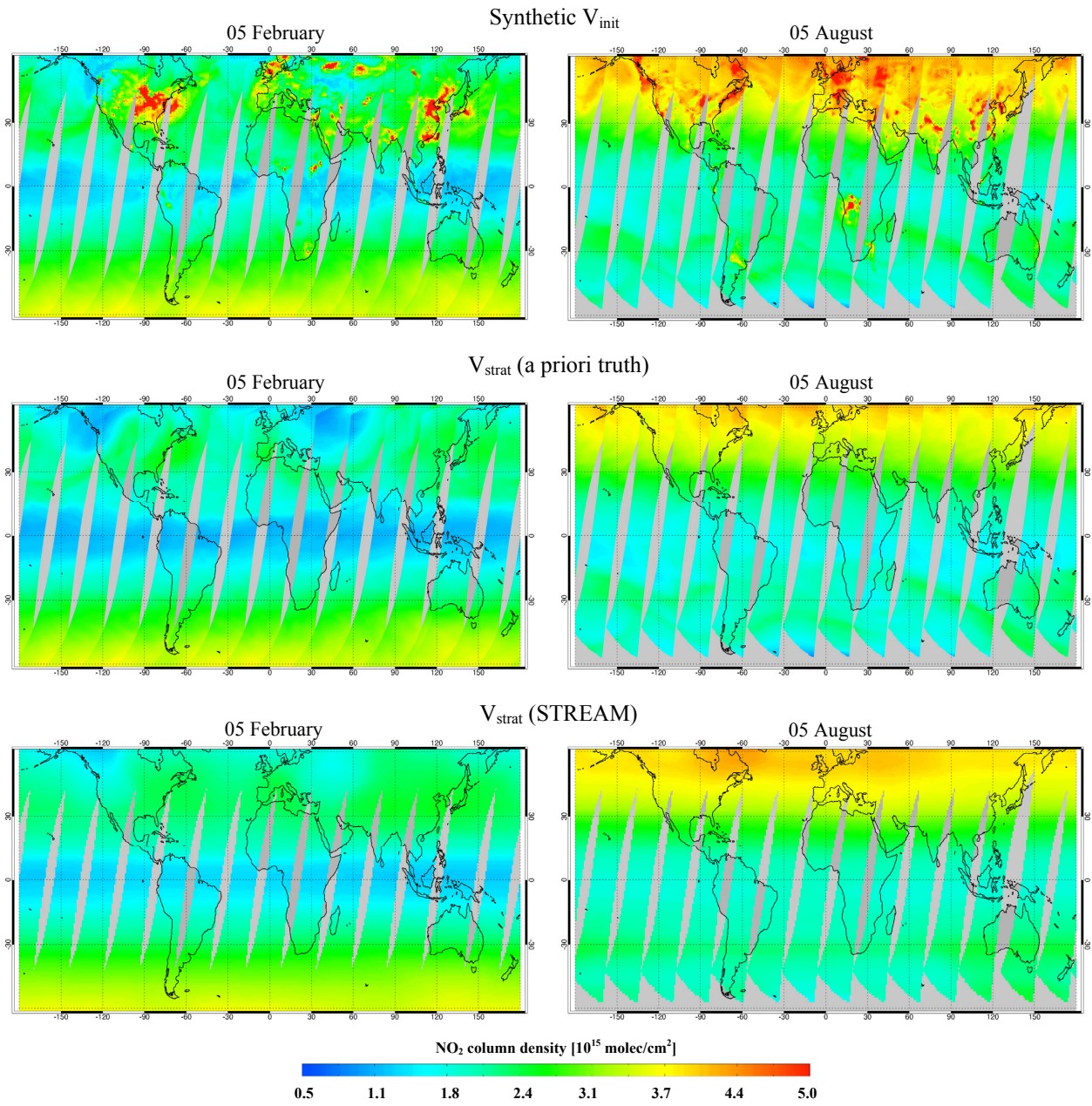

**Figure 7.** Synthetic initial total NO$_2$ columns (top), a priori stratospheric columns from C-IFS (center), and estimated stratospheric columns from STREAM (bottom) on 05 February (left) and 05 August (right) 2009.

by STREAM (Fig. 7 bottom), which is a general limitation of the modified reference sector method. Note that these larger

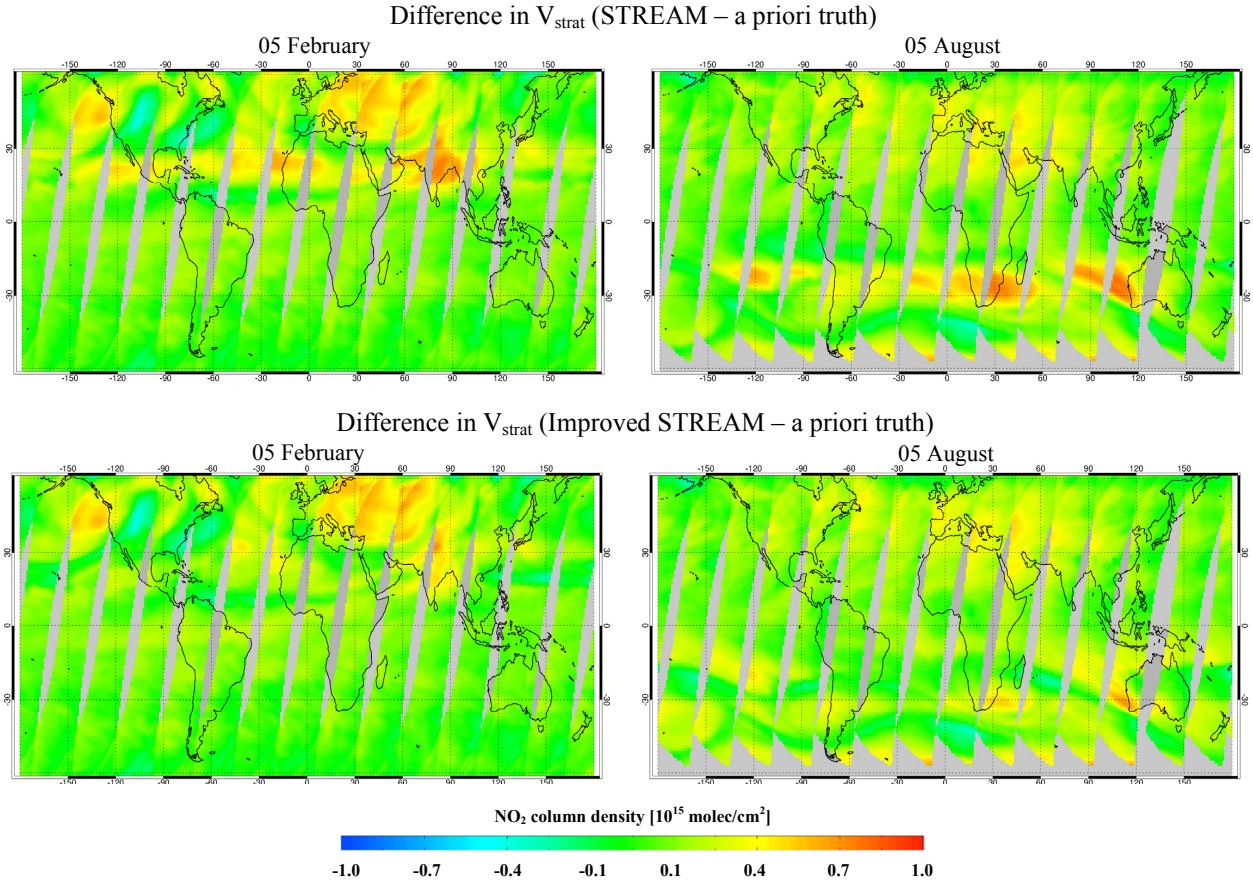

**Figure 8.** Difference in the stratospheric $NO_2$ columns estimated from STREAM and modelled by C-IFS on 5 February (top left) and 5 August (top right) 2009. Bottom panels show for STREAM with improved latitudinal correction.

differences are reduced to $\sim 2 \times 10^{14}$ molec/cm$^2$ in monthly averages (not shown). The found deviations are in agreement with the uncertainty estimates in Beirle et al. (2016).

## 5.2 Improved latitudinal correction

In Fig. 8 (top), larger differences are noticeable over the subtropical regions in winter for both days, primarily related to the latitudinal correction used in STREAM. As described in the previous Sect. 5, the latitudinal correction is applied by determining the latitudinal dependencies of total $NO_2$ over the clean Pacific, removing the latitudinal dependencies before convolution and adding it back to the estimated stratospheric columns. However, longitudinal variations of total $NO_2$, for instance, enhanced total $NO_2$ columns over the Pacific (compared to the Atlantic Ocean) at $15°$N-$30°$N on 5 February 2009 (Fig. 7 top left), introduce biases in the stratospheric $NO_2$ columns. Therefore, an improved latitudinal correction is introduced

**Table 2.** Main settings of AMF calculation method and input data discussed in this study.

|  | GDP 4.8 | GDP 4.9 (this work) |
| --- | --- | --- |
| RTM | LIDORT v2.2+ | VLIDORT v2.7 |
| Surface albedo | TOMS/GOME LER Boersma et al. (2004) | GOME-2 Min-LER v2.1 Tilstra et al. (2017b) |
| A priori profile | Monthly MOZART-2 ($1.875° \times 1.875°$) | Daily TM5-MP ($1° \times 1°$) |

to reduce the biases over the subtropics. The new latitudinal correction determines the latitudinal dependencies of total $NO_2$ based on clean measurements in the whole latitude band (the median of lowest $NO_2$ columns for each $1°$ latitude band). Figure 8 (bottom) shows the difference for the estimated stratospheric $NO_2$ using the improved latitudinal correction. For both days, the application of the new latitudinal correction in STREAM largely removes the biases over the subtropics in Fig. 8 (top).

Applying the improved STREAM on GOME-2 data, Fig. 9 presents the initial total columns from GOME-2 and the strato-spheric $NO_2$ calculated with STREAM and with the spatial filtering method used in the GDP 4.8 algorithm (see Sect. 3) in February and August 2009. For both months, the results calculated with STREAM and with the spatial filtering method show similar global structures. Since the spatial filtering method applies a fixed pollution mask to remove the potentially polluted regions (tropospheric $NO_2$ larger than $1 \times 10^{15}$ molec/cm$^2$), moderately polluted pixels with tropospheric $NO_2$ up

to $1 \times 10^{15}$ molec/cm$^2$ still contribute to the stratospheric estimation. Therefore, enhanced stratospheric $NO_2$ by more than $5 \times 10^{14}$ molec/cm$^2$ is found over polluted regions, e.g., Middle East, China, central Africa, southern Africa, and Australia in Fig. 9 (bottom). This overestimation is largely removed by STREAM in Fig. 9 (center).

# 6   Improvements to $NO_2$ AMF calculation

## 6.1   RTM

As summarized in Table 2, updated box-AMFs are calculated using the linearised vector code VLIDORT (Spurr, 2006) version 2.7. VLIDORT applies the discrete ordinates method to generate simulated intensity and analytic intensity derivatives with respect to atmospheric and surface parameters (i.e. weighting functions). Box-AMFs $m_l$ (see Eq. (3)) are determined as:

$$m_l = \frac{\partial \ln I}{\partial \tau_{NO_2,l}} = \frac{\frac{\partial I}{\partial \tau_{NO_2,l}} \cdot \tau_{NO_2,l}}{I \cdot \tau_{NO_2,l}} \tag{8}$$

with $I$ the simulated top-of-atmosphere radiance, $\tau_{NO_2,l}$ the absorption optical thickness of $NO_2$ at layer $l$, and term $\frac{\partial I}{\partial \tau_{NO_2,l}} \cdot$

$\tau_{NO_2,l}$ the $NO_2$ profile weighting function. Compared to the scalar (intensity-only) LIDORT code, VLIDORT provides more realistic modelling results with a treatment of light polarisation, which affects the tropospheric AMFs by up to 4%.

     The box-AMFs $m_l$ for each layer are calculated for the mid-point wavelength of fitting window, i.e., 461 nm in our $NO_2$ retrieval, which is representative of the window-average box-AMFs. Compared to the tropospheric AMFs at 440 nm (mid-point wavelength in GDP 4.8), the ones calculated at 461 nm are higher by up to 10% for polluted situations, due to the wavelength-

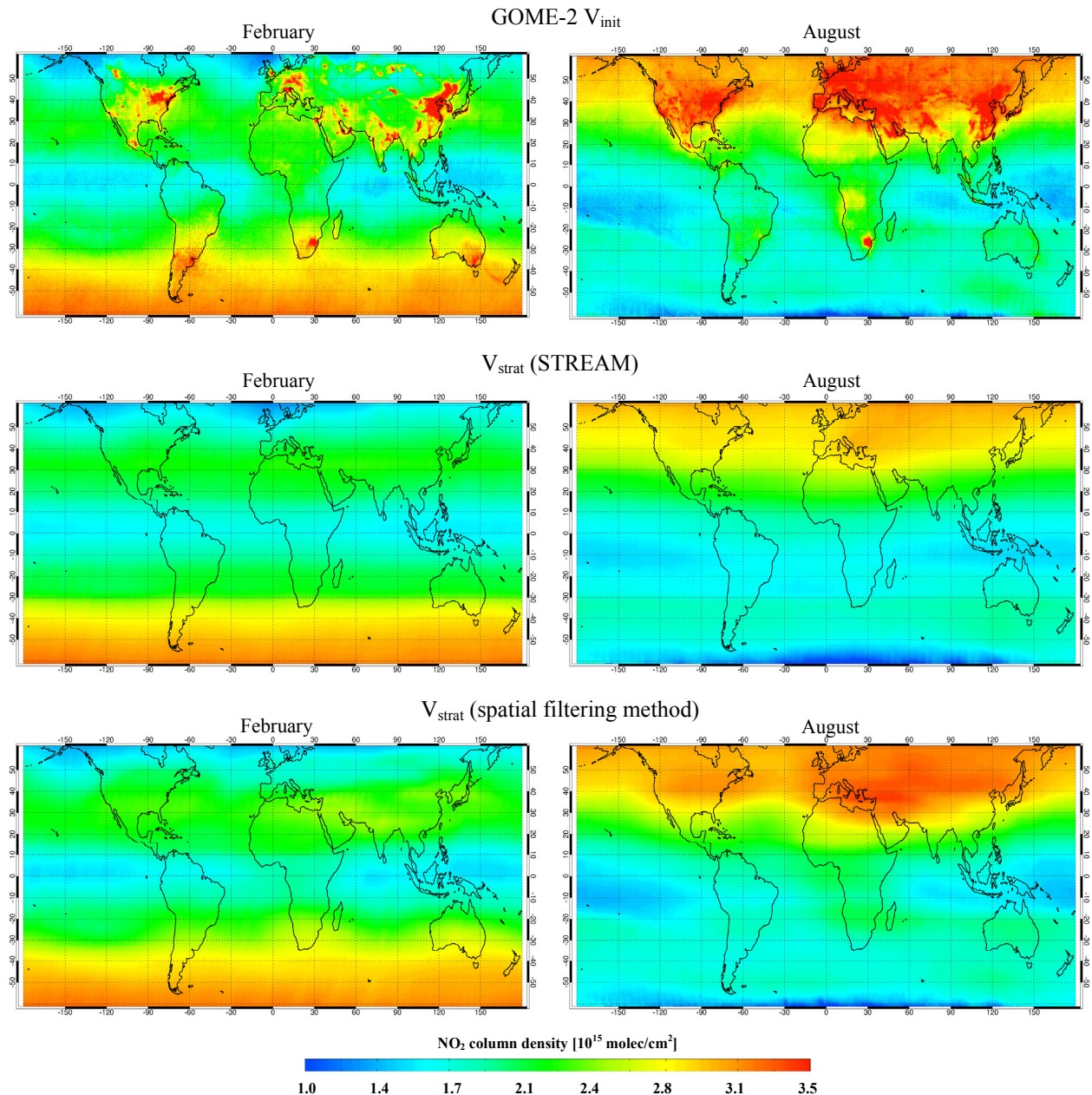

**Figure 9.** GOME-2 initial total $NO_2$ columns (top) and stratospheric $NO_2$ columns retrieved from the improved STREAM algorithm (center) and from the spatial filtering method used in GDP 4.8 (bottom), measured by GOME-2A in February (left) and August (right) 2009.

dependency of Rayleigh scattering, in agreement with Boersma et al. (2018) (see Fig 7 therein). Note that the uncertainty

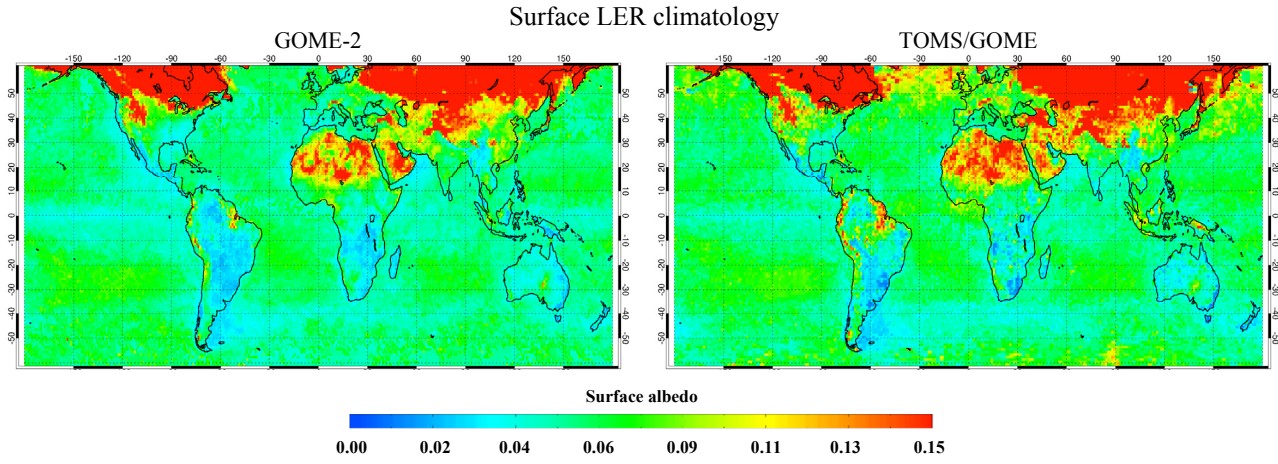

**Figure 10.** Map of surface LER data for 440 nm in February based on GOME-2 observations for 2007-2013 (Tilstra et al., 2017b) (left) and TOMS/GOME data for 1979-1993 (right).

related to the wavelength-dependency of the AMF is much smaller than the uncertainties introduced by surface albedo, a priori $NO_2$ profile, cloud and aerosol (see Sect. 6.4).

$m_l$ is calculated with the RTM and stored in a LUT as a function of GOME-2 viewing geometry, surface pressure, and surface albedo. Compared to the LUT used in the GDP 4.8, a new LUT is calculated with an increased number of reference points, e.g., for surface pressure (from 10 to 16) and for surface albedo (from 10 to 14), as well as vertical layers (from 24 to 60) to reduce the interpolation error (Lorente et al., 2017), leading to differences in tropospheric AMFs by up to 2%.

## 6.2 Surface albedo

Surface albedo is an important parameter for an accurate retrieval of $NO_2$ columns and cloud properties. The sensitivity of backscattered radiance to the boundary layer $NO_2$ is strongly related to the surface albedo, especially over polluted areas. In the GDP 4.9, the surface LER climatology based on TOMS/GOME data (Boersma et al., 2004) has been replaced by one based on GOME-2 observations (Tilstra et al., 2017b). Using the degradation-corrected GOME-2 level 1 measurements, the GOME-2 surface LER is derived by matching the measurements in a pure Rayleigh scattering atmosphere without cloud. Compared to the TOMS/GOME LER climatology, the GOME-2 surface LER (version 2.1) dataset takes advantage of newer observations for 2007-2013, an increased spatial resolution of $1.0°$lon$×1.0°$lat for standard grid cells and $0.25°$lon$×0.25°$at coastlines (Tilstra et al., 2017a), and an improved treatment of cloud contaminated cells over the ocean.

Figure 10 shows the surface LER data from the GOME-2 and TOMS/GOME observations for 440 nm in February. A good overall consistency is found between the two LER datasets, in particular over the ocean. Larger differences are found over certain snow or ice areas, like Russia and southern Canada, which can be attributed to changes in snow or ice cover during the different measurement periods of the two LER datasets. Increased spatial resolution for the GOME-2 LER version 2.1 dataset

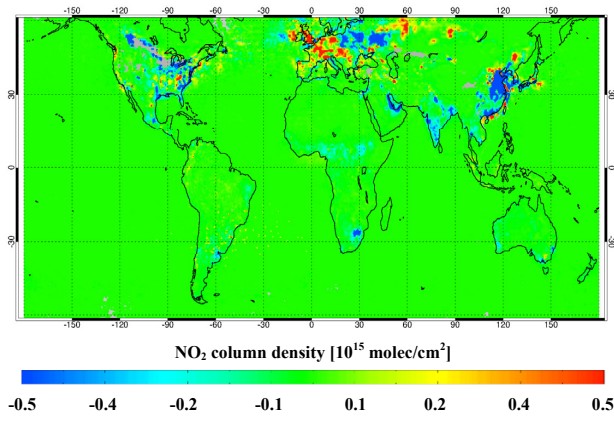

**Figure 11.** Difference in tropospheric $NO_2$ columns for clear-sky conditions (cloud radiance fraction smaller than 0.5) for February 2008 retrieved using the GOME-2 surface LER climatology version 2.1 and the LER climatology based on TOMS/GOME data at 440 nm.

enables a better representation of surface features for the land-sea boundaries, e.g., coasts around western Europe and eastern China. Improvements in the GOME-2 LER algorithm (Tilstra et al., 2017b) decreases the surface LER values over regions with persistent clouds, e.g., the North Atlantic Ocean and the North Pacific Ocean at middle latitudes. Systematic differences in the LER climatologies are also caused by the different overpass time, observing geometry, and radiometric calibration of the instruments.

Figure 11 illustrates the influence of the updated surface LER at 440 nm on the retrieved tropospheric $NO_2$ columns in February 2008. The difference over the ocean is very small. Larger effects are noticed primarily under polluted conditions with positive differences, e.g., over parts of central Europe, Russia or USA, and negative values, e.g., over parts of South Africa, India or China. The differences in the retrieved tropospheric $NO_2$ columns are consistent with the changes in the surface LER. For example, the GOME-2 surface LER over central Europe is $\sim$0.012 smaller than TOMS/GOME data, and a lower sensitivity to tropospheric $NO_2$ is therefore assumed in the AMF calculation. This results in a decrease in the AMF and hence an increase in the retrieved tropospheric $NO_2$ column by $\sim 7 \times 10^{14}$ molec/cm$^2$ ($\sim$12%). Vice versa, an increase of the surface LER values by $\sim$0.018 over the Yangtze River region in eastern China leads to a reduction of tropospheric $NO_2$ columns by $\sim 4 \times 10^{15}$ molec/cm$^2$ ($\sim$15%).

As described in Sect. 6.1, the AMFs are calculated for 461 nm in the GDP 4.9 (425-497 nm wavelength window) instead of 440 nm in the GDP 4.8 (with 425-450 nm wavelength window), therefore the corresponding surface LER values are 463 nm are used. The surface LER values at 463 nm are higher by up to 0.02 over desert areas and lower by up to 0.02 over the ocean and the snow or ice areas, which result in differences of up to 5% in the calculated AMFs.

The surface LER climatology from Kleipool et al. (2008) derived from OMI measurements for 2004-2007 has been widely used in satellite $NO_2$ retrievals (e.g., Boersma et al., 2011; Barkley et al., 2013; Bucsela et al., 2013). An important advantage of using the GOME-2 LER climatology with respect to the OMI LER dataset in our retrieval is the consistency with the GOME-2

NO$_2$ observations, considering the illumination conditions, observation geometry, and instrumental characteristics. Another advantage of the GOME-2 LER climatology is the use of more recent observations to reduce the errors introduced by ignoring the interannual variability of surface albedo, which are possibly large for varying snow and ice situations. Possible corrections for the surface albedo from a climatology include the use of external information about the actual snow and ice conditions,
e.g., from Near-real-time Ice and Snow Extent (NISE) dataset (Nolin et al., 2005).

## 6.3   A priori vertical profiles

The retrieved tropospheric NO$_2$ columns are sensitive to changes in the relative vertical distribution of the a priori NO$_2$ concentrations (i.e. profile shape). Increasing the spatial and/or temporal resolution of the a priori profiles have shown to produce a more accurate NO$_2$ retrieval (e.g., Russell et al., 2011; Heckel et al., 2011; McLinden et al., 2014; Nüß et al., 2006; Laugh-
ner et al., 2016). To improve the tropospheric AMF calculation, daily a priori NO$_2$ profiles are obtained with a resolution of 1°lon×1°lat from the chemical transport model TM5-MP (Huijnen et al., 2010; Williams et al., 2017). The TM5-MP profiles have been used in several studies to derive AMFs and tropospheric NO$_2$ columns (e.g., van Geffen et al., 2016; Lorente et al., 2017; Boersma et al., 2018).

Figure 12 shows the TM5-MP and MOZART-2 a priori NO$_2$ profiles for two pollution hot spots located in Brussels (Belgium,
50.9°N, 4.4°E) and Guangzhou (China, 23.1°N, 113.3°E) on one day in February and August 2009 as examples. Monthly profiles are shown for MOZART-2, and profiles for the given days are shown for TM5-MP. Large differences between the a priori NO$_2$ profile shapes from TM5-MP and MOZART-2 are found for both cities. These differences are the result of the different chemical mechanism, transport scheme, and emission inventory employed by the model, the different spatial resolution, and the use of daily vs. monthly profiles. In TM5-MP, the use of updated NO$_x$ emissions from the MACCity
inventory (Granier et al., 2011) produces more realistic profiles. Improvement in the spatial resolution gives a more accurate description of the NO$_2$ gradient and transport. The use of daily profiles provides a better description of the temporal NO$_2$ variation, especially for regions dominated by emission and transport like Brussels and Guangzhou.

In Fig. 12, the tropospheric NO$_2$ columns retrieved for the individual days using TM5-MP and MOZART-2 a priori NO$_2$ profiles are also reported. Taking Brussels on 11 February 2009 (Fig. 12 top left) as an example, the smaller boundary layer
concentration modelled by TM5-MP (less steep profile shape) leads to an increase in the tropospheric AMF and hence a decrease in the retrieved tropospheric NO$_2$ columns by $2.6 \times 10^{15}$ molec/cm$^2$ (19.7%). Figure 13 presents a comparison of the monthly averaged tropospheric NO$_2$ columns retrieved using daily TM5-MP and monthly MOZART-2 a priori NO$_2$ profiles in February and August 2009. The application of the daily TM5-MP a priori NO$_2$ profiles affects the tropospheric NO$_2$ columns by more than $1 \times 10^{15}$ molec/cm$^2$ mostly over polluted regions with enhanced NO$_2$ in the boundary layer, e.g., with an increase
of tropospheric NO$_2$ over parts of China, India, and South Africa, and a decrease over parts of eastern US, Europe, and Japan.

To analyse the effect of using daily vs. monthly profiles, the tropospheric NO$_2$ columns are also retrieved using the monthly average TM5-MP profiles, as shown in Fig. 12. Differences in the profile shape of daily and monthly profiles are mainly related to the variations in the meteorology. In agreement with Nüß et al. (2006) and Laughner et al. (2016), the use of monthly profiles changes the tropospheric NO$_2$ columns by up to $3 \times 10^{15}$ molec/cm$^2$ depending on the wind speed and wind direction,

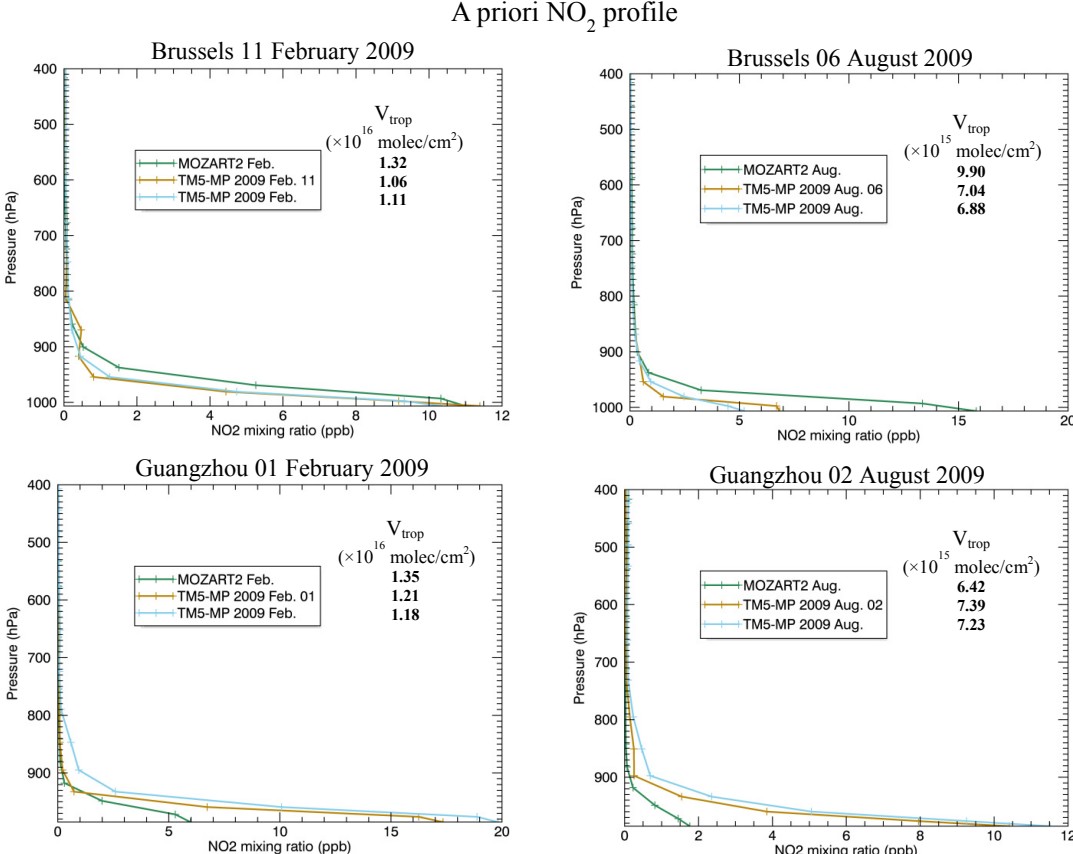

**Figure 12.** Examples of a priori NO$_2$ profiles for Brussels (top) and Guangzhou (bottom) on a given day in February (left) and August (right) 2009. Monthly profiles are shown for MOZART-2 (green), and daily profiles on the given days are shown for TM5-MP (brown) together with the monthly average profiles calculated for TM5-MP (blue). The tropospheric NO$_2$ columns retrieved using each a priori NO$_2$ profile are also given.

in particular for regions affected by transport (not shown). For the example of Brussels on 11 February 2009 (Fig. 12 top left), the use of monthly profiles increases the tropospheric NO$_2$ columns by $5 \times 10^{14}$ molec/cm$^2$ (4.7%). A comprehensive analyse of the effect of using a priori NO$_2$ profiles from different chemistry transport models on the retrieved tropospheric NO$_2$ will be described in a subsequent paper.

5 **6.4 Examples of GOME-2 tropospheric NO$_2$**

Figure 14 shows the tropospheric NO$_2$ columns from the improved GDP 4.9 algorithm for February and August averaged for the year 2007-2016. Figure 15 shows the difference in tropospheric NO$_2$ columns from the GDP 4.9 and GDP 4.8 product. The tropospheric NO$_2$ columns increase globally by $\sim 1 \times 10^{14}$ molec/cm$^2$ due to the improved DOAS slant column fitting

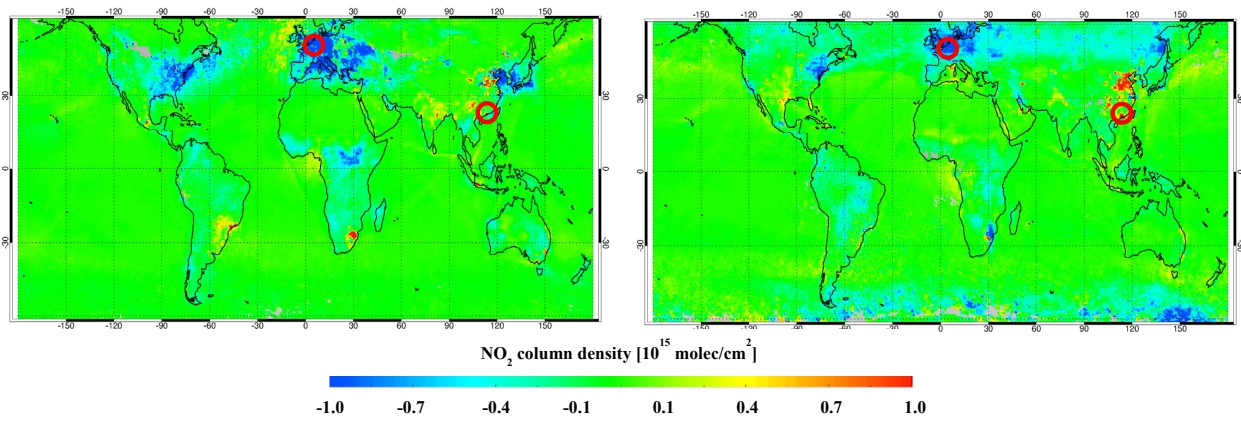

**Figure 13.** Difference in tropospheric NO$_2$ columns for clear-sky conditions (cloud radiance fraction smaller than 0.5) retrieved using daily TM5-MP and monthly MOZART-2 a priori NO$_2$ profiles for February (left) and August (right) 2009. Red circles indicate locations in Fig. 12.

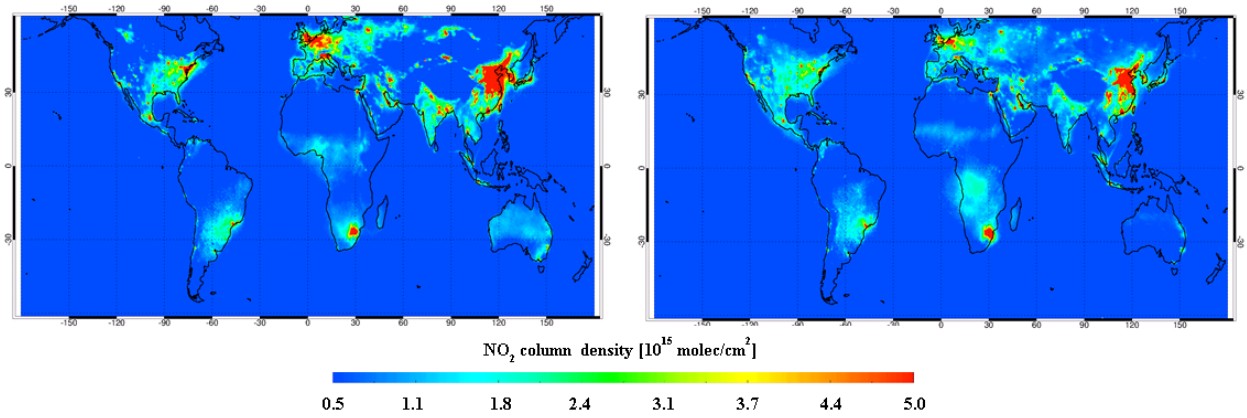

**Figure 14.** Monthly average tropospheric NO$_2$ columns from GDP 4.9 for clear-sky conditions (cloud radiance fraction smaller than 0.5), measured by GOME-2A in February (left) and August (right) 2007-2016.

and increase further by $\sim$$3 \times 10^{14}$ molec/cm$^2$ around moderately polluted regions beneficial from the use of new stratosphere-troposphere separation algorithm STREAM. A stronger change by more than $1 \times 10^{15}$ molec/cm$^2$ is found mainly over polluted continents, as a result of the improvements to the AMF calculation, primarily the surface albedo (which also affects the snow or ice area, e.g., southern Canada and northeastern Europe) and/or the a priori NO$_2$ profiles (which also affects the polluted ocean, e.g., shipping lanes in southeastern Asia).

Over central northern Europe, the tropospheric NO$_2$ columns are reduced by $\sim$$1 \times 10^{15}$ molec/cm$^2$ for GDP 4.9 in winter and $\sim$$3 \times 10^{14}$ molec/cm$^2$ in summer. A larger number of negative values in GDP 4.8, possibly related to the overestimated

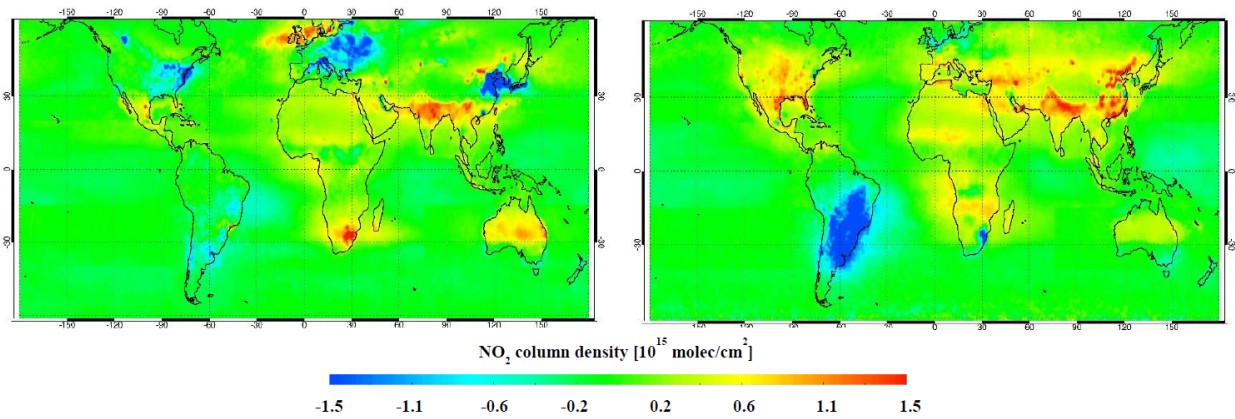

**Figure 15.** Difference in tropospheric $NO_2$ columns from GDP 4.9 and GDP 4.8 for clear-sky conditions (cloud radiance fraction smaller than 0.5) in February (left) and August (right) 2007-2016 for GOME-2A.

stratospheric $NO_2$ around polar vortex areas, is largely corrected in GDP 4.9 by improving the stratosphere-troposphere separation algorithm. Over eastern China and eastern US, the seasonal variation is consistent between GDP 4.8 and 4.9, with reduced values in winter (by more than $1 \times 10^{15}$ molec/cm$^2$) and enlarged values in summer (by more than $1 \times 10^{15}$ molec/cm$^2$ for eastern China and $5 \times 10^{14}$ molec/cm$^2$ for eastern US) for GDP 4.9 due to the combined impact of the algorithm changes, mainly the AMF calculation. Over India and its surrounding areas, a systematic increase in tropospheric $NO_2$ columns by $\sim 7 \times 10^{14}$ molec/cm$^2$ for GDP 4.9 benefits from the use of STREAM.

### 6.5 Uncertainty estimates for GOME-2 total and tropospheric $NO_2$

The uncertainty in our GDP 4.9 $NO_2$ slant columns is $4.4 \times 10^{14}$ molec/cm$^2$, calculated from the average slant column error using a statistical method described in Sect. 4.5. The uncertainty in the GOME-2 stratospheric columns is $\sim$4-5 $\times 10^{14}$ molec/cm$^2$ for polluted conditions based on the daily synthetic GOME-2 data and $\sim$1-2 $\times 10^{14}$ molec/cm$^2$ for monthly averages. The uncertainty in the GDP 4.9 AMF calculation is likely reduced, considering the improved surface albedo climatology and a priori $NO_2$ profiles, which are the main causes of AMF structural uncertainty (Lorente et al., 2017). In addition, the AMF uncertainty is substantially driven by the cloud parameters and the aerosol correction approach.

The largest cloud-related uncertainty in $NO_2$ retrieval is introduced by the surface albedo-cloud fraction error correlation, as analysed by Boersma et al. (2018) for OMI using OMCLDO2 cloud product, which requires a surface albedo climatology as input in the cloud fraction retrieval. But this uncertainty is likely smaller for OCRA/ROCINN cloud algorithms, since the surface albedo is treated differently in OCRA's cloud fraction calculation. Retrieved by separating a spectral scene into cloudy contribution and cloud-free background, the cloud fraction from OCRA is affected by surface albedo through the cloud-free map construction with a larger impact over bright surfaces like snow or ice cover, in particular during snowfall (higher

background) or melting (lower background), which has been corrected by interpolating towards a daily value between two monthly cloud-free map in OCRA (Lutz et al., 2016).

The uncertainty introduced by aerosol in GDP 4.9 is ∼50% for high aerosol loading, in agreement with Lorente et al. (2017). With direct impact on $NO_2$ AMF calculation and indirect impact via cloud parameters retrieval, the aerosol effect has been

considered for OMI implicitly through the cloud correction (Boersma et al., 2004, 2011) or explicitly with additional aerosol information for regional studies (Lin et al., 2014, 2015; Kuhlmann et al., 2015; Castellanos et al., 2015; Chimot et al., 2018), leading to an increase or decrease of $NO_2$ AMF by up to 40% depending on $NO_2$ distribution and aerosol properties and distribution. Since aerosol is highly variable in space and time due to the dependency on emission sources, transports, and atmospheric processes (Holben et al., 1991), explicit aerosol correction will be applied in our AMF calculation when reliable

observations or model outputs of aerosol optical properties and vertical distributions are available. To conclude, the uncertainty in the AMF calculation is estimated to be in the 10-45% range for polluted conditions, leading to a total uncertainty in the tropospheric $NO_2$ columns likely in the range of 30-70%.

## 7 End-to-end GOME-2 $NO_2$ validation

The validation of $NO_2$ data derived from GOME-2 GDP algorithm is part of the validation activities done at BIRA-IASB in

the AC-SAF context (Hassinen et al., 2016). An end-to-end validation approach is usually performed for each main release and summarized in validation reports that can be found on AC-SAF validation website (http://cdop.aeronomie.be/validation/ valid-reports). This includes several steps, such as: (1) the DOAS analysis results, cloud properties retrievals, and AMF evaluations by confrontation of GOME-2 retrievals to other established satellite retrievals and AMF evaluations; (2) the stratospheric reference evaluation by comparison with correlative observations from ground-based zenith-looking DOAS spectrometers and

from other nadir-looking satellites; and (3) the tropospheric and total $NO_2$ column data evaluation by comparison with correlative observations from ground-based multiple-axis DOAS (MAXDOAS) and Direct Sun spectrometers (Pinardi et al., 2014). In this paper, we focus on the last point: the validation of tropospheric data with BIRA-IASB ground-based MAXDOAS data.

The MAXDOAS instruments collect scattered sky light in a series of line-of-sight angular directions extending from the horizon to the zenith. High sensitivity towards absorbers near the surface is obtained for the smallest elevation angles, while

measurements at higher elevations provide information on the rest of the column. This technique allows the determination of vertically resolved abundances of atmospheric trace species in lowermost troposphere (Hönninger et al., 2004; Wagner et al., 2004; Wittrock et al., 2004; Heckel et al., 2005). Here the bePRO retrieval code (Clémer et al., 2010; Hendrick et al., 2014; Vlemmix et al., 2015) is used to retrieve tropospheric columns and low tropospheric profiles (up to 3.5 km with about 2 to 3 degrees of freedom).

As summarised in Table 3, a set of MAXDOAS stations (Beijing, Bujumbura, Observatoire de Haute Provence (OHP), Reunion, Uccle, and Xianghe) is providing interesting test cases for GOME-2 sensitivity to tropospheric $NO_2$. Indeed Beijing and Uccle are typical urban stations, Xianghe is a suburban station (∼60 km from Beijing), Bujumbura and Reunion are small cities in remote regions, and OHP is largely rural but occasionally influenced by polluted air masses transported from

**Table 3.** An overview of BIRA-IASB MAXDOAS datasets used in this study.

| MAXDOAS station | period | position | description |
| --- | --- | --- | --- |
| Beijing | 6/2008-4/2009 | 39.98°N, 116.38°E | urban polluted site in China |
| Bujumbura | 12/2013-11/2016 | 3.38°S, 29.38°E | urban site in Burundi |
| OHP | 3/2007-11/2016 | 43.94°N, 5.71°E | background site in southern France |
| Reunion | 4/2016-11/2016 | 21°S, 55.3°E | urban site in Reunion Island |
| Uccle | 4/2011-11/2016 | 51°N,4.36°E | urban polluted site in Belgium with a miniDOAS |
| Xianghe | 3/2010-11/2016 | 39.75°N, 116.96°E | suburban polluted site in China |

neighboring cities. These different station types are important in the validation context as it is generally expected that urban stations are underestimated by the satellite data, due to the averaging of a local source over a pixel size ($80\times40/40\times40$ km$^2$ for GOME-2) larger than the horizontal sensitivity of the ground-based measurements which is about few to tens of km (Irie et al., 2011; Wagner et al., 2011; Ortega et al., 2015). In this context, MAXDOAS data is already better than in-situ
measurements with an extended horizontal and vertical sensitivity, more similar to the satellite sensitivity, but differences in sampling and sensitivity still remain and explain part of the biases highlighted by validation exercises. Several validation studies show significant underestimation of tropospheric trace gases, such as NO$_2$, from satellite observations over regions with strong spatial gradients in tropospheric pollution (e.g., Celarier et al., 2008; Kramer et al., 2008; Chen et al., 2009; Irie et al., 2012; Ma et al., 2013; Wu et al., 2013; Kanaya et al., 2014; Wang et al., 2017; Drosoglou et al., 2017, 2018). Other possible
explanations include the uncertainties in the applied satellite retrieval assumptions, such as the choices of surface albedo, a priori NO$_2$ profiles, or cloud and aerosol treatment (Boersma et al., 2004, 2011; Leitão et al., 2010; Heckel et al., 2011; Lin et al., 2014, 2015). The best agreement is generally obtained in the case of suburban and remote stations, but difficulties may arise when small local sources are present in a remote location, such as Reunion Island or Bujumbura (Pinardi et al., 2015; Gielen et al., 2017).

The same methodology as in the GDP 4.8 validation report (Pinardi et al., 2015) is used for the validation of this improved GDP 4.9 tropospheric NO$_2$ dataset: the satellite data are filtered for clouds (cloud radiance fraction smaller than 0.5) and the mean value of all the valid pixels within 50 km of the stations is compared to the ground-based value. The original ground-based MAXDOAS data usually retrieves NO$_2$ columns all day long every 20 to 30 minutes, and these values are linearly interpolated to the GOME-2 overpass time (9:30 local time), if original data exist within +/-1 hours.

Figure 16 shows an example of the time-series and scatter plot of the daily and monthly means comparison between GDP 4.9 tropospheric NO$_2$ columns and ground-based MAXDOAS measurements in Xianghe, including the statistical information on the number of points, correlation coefficient, slope and intercept of orthogonal regression analysis. Figure 17 presents the daily and monthly mean absolute and relative differences of GDP 4.9 and ground-based measurements. As can be seen in Fig. 16 and 17, the seasonal variation in the tropospheric NO$_2$ columns is similarly captured by both observation systems with differences
on average within $\pm3 \times 10^{15}$ molec/cm$^2$ (median difference of $-1.2 \times 10^{15}$ molec/cm$^2$). Larger differences are observed on

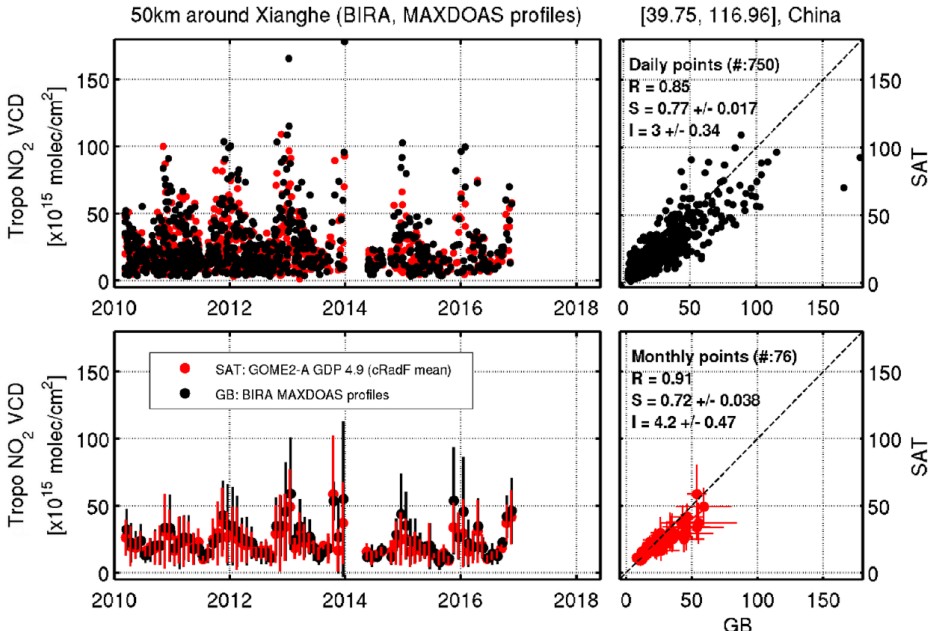

**Figure 16.** Daily (upper row) and monthly mean (lower row) time series and scatter plot of GOME-2A and MAXDOAS tropospheric $NO_2$ columns (mean value of all the pixels within 50km around Xianghe).

some days and months, in particular in winter when $NO_2$ and aerosol loadings are large. A relatively compact scatter is found, with a correlation coefficient of 0.91 and a slope of 0.72±0.04 for the orthogonal regression fit. These results are qualitatively similar to those obtained in previous validation exercises (Celarier et al., 2008; Kramer et al., 2008; Chen et al., 2009; Irie et al., 2012; Ma et al., 2013; Wu et al., 2013; Kanaya et al., 2014; Wang et al., 2017; Drosoglou et al., 2017, 2018). Similar figures for GDP 4.8 can be found on the AC-SAF validation website (http://cdop.aeronomie.be/validation/valid-results).

Figure 18 reports the monthly mean absolute and relative differences for both GDP 4.8 and GDP 4.9 for Xianghe station. The daily differences are also reported through the histogram panel, where the reduction in the spread of the daily comparison points is clearly visible for GDP 4.9. The reduction of the bias, which is smaller and more stable in time, is seen in the absolute and relative monthly mean bias time-series. Three years show a standard deviation of the monthly biases larger for GDP 4.9 than for GDP 4.8 (±12% instead of ±8% in 2010, ±12% instead of ±8% in 2013, and ±41% instead of ±27% in 2014) but with a strongly reduced mean bias (-4% instead of -20%, -8% instead of -34%, and -1% instead of -44%).

Similar figures as Fig. 16 and 18 for all the stations are gathered in Fig. S1 to S4 in the supplement, and all the statistics are summarized in Table 4 and Table 5 for GOME-2A and GOME-2B, respectively. Fig. S1 and S2 present the time-series and scatter plots for GDP 4.9, while Fig. S3 and S4 present the differences for both GDP 4.9 and GDP 4.8 comparisons. As discussed in Pinardi et al. (2015), for background stations (here Bujumbura, Reunion and OHP), the mean bias is considered

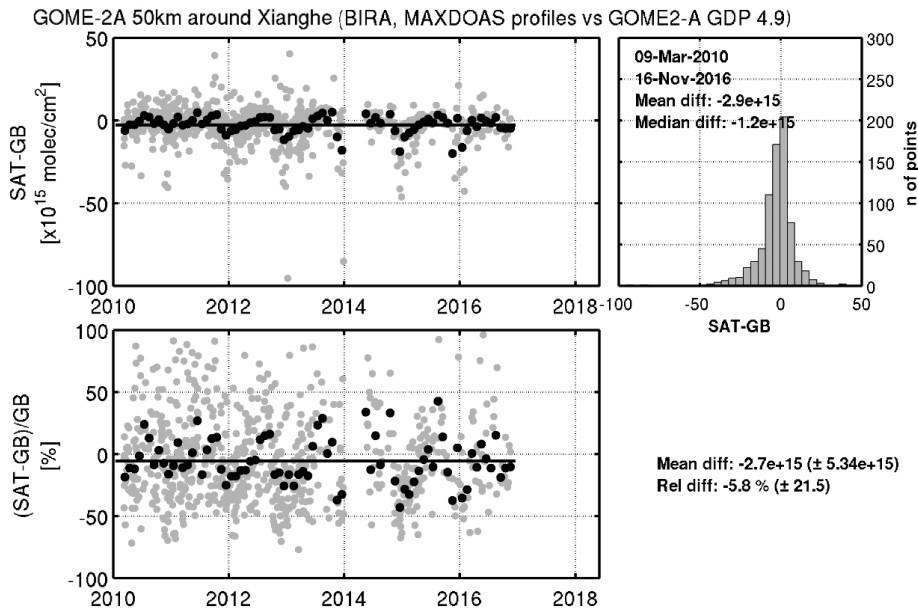

**Figure 17.** Daily (grey dots) and monthly mean (back dots) absolute and relative GOME-2A and MAXDOAS time series differences for the Xianghe station. The histogram of the daily differences is also given, with the mean and median difference, and the total time-series absolute and relative monthly differences are given outside the panels.

as the best indicator of the validation results, due to the relatively small variability in the measured $NO_2$. In urban (Beijing and Uccle) and suburban (Xianghe) situations, the $NO_2$ variability is large enough and in this case, the correlation coefficient is a good indication of the linearity or coherence of the satellite and ground-based dataset, although larger difference in term of slope (closer to 0.5 than to 1 for urban cases) and of mean bias can be expected because satellite measurements (and especially

GOME-2 $80{\times}40/40{\times}40$ km$^2$ pixels) smooth out the local $NO_2$ hot spots. This can be seen e.g. in the cases of Beijing and Xianghe for GOME-2A (see Fig. S1a and Fig. 16, respectively), where very high correlation (R=0.94 and 0.91, respectively) are obtained from GDP 4.9, showing the very consistent behavior of both datasets for small and large $NO_2$ columns, while their slopes (S=0.4 and 0.72, respectively) show almost a factor 2 of difference, with a smaller slope in the Beijing case, where the MAXDOAS instrument is in the city center and thus much more subject to local emission smeared out by the GOME-2

large pixel. This last effect is also seen through the biases values (RD=-47% and -5.8%, respectively) that are strongly reduced when moving the MAXDOAS outside the city in a suburban location like Xianghe. Slope of 0.47 (similar to the 0.4 of Beijing) is also obtained in Uccle, another urban site, where the MAXDOAS is affected by local emissions.

In remote cases such as OHP, Bujumbura or Reunion Island, as discussed above, the variation of the $NO_2$ columns are small and the statistical analysis on the regression are not very representative of the situation, with a cloud of points giving

small slopes and low correlation coefficients (see e.g. Fig. S1b to d and Table 4 for GOME-2A). In those cases, GOME-2 is

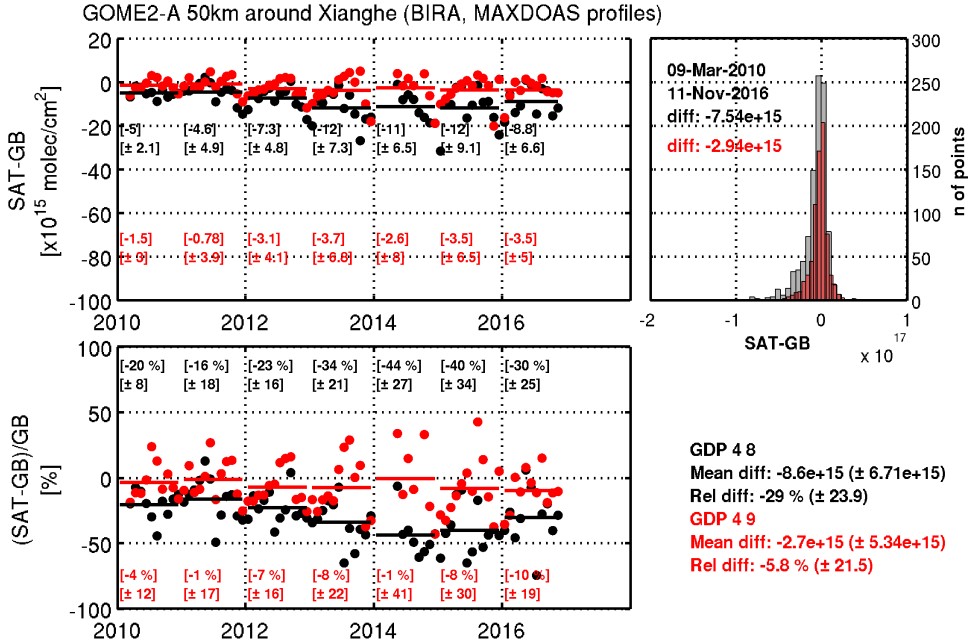

**Figure 18.** Absolute and relative differences of GOME-2A and MAXDOAS tropospheric $NO_2$ columns. The time-series presents the monthly mean differences for GDP 4.8 (black) and GDP 4.9 (red). The total mean differences values and standard deviations are given, as well as the yearly values. The histogram presents the daily differences over the whole time-series for the two products (grey for GDP 4.8 and red for GDP 4.9).

lower than the ground-based, with sometime almost no seasonal variation, e.g. Bujumbura and Reunion, and in other cases, like OHP, some of the daily peaks are captured by GOME-2 (as days in the winter of 2014 and 2015), and the seasonal patterns and the orders of magnitudes of both datasets are similar. In these cases, it is best to look at the absolute biases (as relative biases are large due to the division with small ground-based columns), as presented in e.g. Fig. S3b to d and Table 4.

5     Mean absolute differences for GDP 4.9 are about $-3.6 \times 10^{15}$ molec/cm$^2$ for Bujumbura, $-8.5 \times 10^{14}$ molec/cm$^2$ for OHP, and $-1.5 \times 10^{15}$ molec/cm$^2$ for Reunion, which are all smaller than their respective GDP 4.8 values. The daily differences presented in the histograms of those figures also show reduced spread of GDP 4.9 comparisons when superposed to the GDP 4.8 results. Similar differences are also found for GOME-2B.

     To conclude, although the Xianghe case presented in Fig. 16 to 18 is the best case (due to its suburban location and its

10    long time-series), a better seasonal agreement between GDP 4.9 and MAXDOAS data is found for urban and suburban cases like Beijing, Uccle, and Xianghe, compared to results with GDP 4.8. In remote locations such as OHP, which is occasionally influenced by polluted air masses transported from neighboring cities, the comparison is also meaningful (e.g. with a mean

bias reduced from -45% for GDP 4.8 to -25% for GDP 4.9 for GOME-2A), while cases such as Bujumbura and Reunion are quite challenging for satellite validation, with specific local conditions (Bujumbura is in a valley on the side of the Tanganyika lake, while the MAXDOAS at Reunion is in St-Denis, on the coast of the 65 km long and 50 km wide island in the Indian ocean, containing a mountain massif with summits above 2740 m asl). In both cases the MAXDOAS instrument is located in small cities surrounded by specific orography, difficult for satellite retrievals and challenging for validation. The absolute and relative differences show, however, a clear improvement for all the stations, when comparing to GDP 4.8 results for both daily and monthly mean biases. The daily biases and spreads are all reduced.

To summarize, the impact of the improvement of the algorithm (as seen in Table 4 and 5 and in Fig. S3 and S4) leads to a decrease of the relative differences in urban conditions such as in Beijing or Uccle from [-52,-60]% for GDP 4.8 to [-43,-47]% for GDP 4.9 for GOME-2A and from -54% to -40% for GOME-2B. In suburban conditions such as in Xianghe, the differences go from -30% to -6% for GOME-2A and from -26% to -2% for GOME-2B. In remote (difficult) cases such as in Bujumbura or Reunion, the differences go from [-89,-90]% to [-64,-76]% for GOME-2A and from [-86,-87]% to [-47,-74]% for GOME-2B, while in background case such as in OHP, the differences decrease from -45% to -25% for GOME-2A and from -42% to -17% for GOME-2B. The differences in numbers for GOME-2A and GOME-2B are due to the different time-series length of both comparisons (e.g. March 2010-November 2016 for GOME-2A and December 2012-November 2016 for GOME-2B in Xianghe), the different sampling of the atmosphere by GOME-2A and GOME-2B (slight time-delay between both overpasses and reduced swath pixels for GOME-2A since July 2013), and the impact of the decreasing quality of the satellite in time, i.e. the GOME-2A degradation (Dikty et al., 2011; Munro et al., 2016). This lead, e.g. for Xianghe, to -2% bias and 0.49 slope for GOME-2B compared to -6% and 0.72 for GOME-2A for GDP 4.9.

These comparisons results aim at showing how the final GDP 4.9 product is improved compared to its predecessor, and not to summarize the improvements of each of the changes discussed in previous sections. In addition, the specific validation method could be improved or at least better characterized (including results uncertainties), by e.g. changing the colocation method (averaging the MAXDOAS within an hour of the satellite overpass or selecting the closest satellite pixel, or only considering the pixels containing the station, etc.), but this is out of the scope of the present manuscript that wants to compare to standard validation results performed routinely on GDP 4.8 (and publicly available on http://cdop.aeronomie.be/validation/valid-results).

For most stations, in addition of the tropospheric columns, MAXDOAS retrieved $NO_2$ profiles can also be exploited with satellite column averaging kernels (AK) to further investigate the impact of the satellite a priori $NO_2$ profiles in the comparison differences (Eskes and Boersma, 2003). The satellite AK describes the vertical sensitivity of measurements to $NO_2$ concentrations and relates the MAXDOAS profiles to satellite column measurements by calculating the "smoothed MAXDOAS columns" as:

$$V_{MAXDOAS,smoothed} = \sum_l AK_{sat,l} \times x_{MAXDOAS,l}. \tag{9}$$

The smoothed MAXDOAS $NO_2$ columns $V_{MAXDOAS,smoothed}$ are derived for each day by convolving the layer ($l$)-dependent daily profile (interpolated to the satellite overpass time) $x_{MAXDOAS}$ expressed in partial columns with the satellite column averaging kernel $AK_{sat}$.

**Table 4.** Averaged Absolute Differences (AD, SAT-GB in $10^{15}$ molec/cm$^2$), Relative Differences (RD, (SAT-GB)/GB in %), standard deviation (STDEV), correlation coefficient R and regression parameters (slope S and intercept I) of the orthogonal regression for the monthly means GOME-2A tropospheric NO$_2$ product when comparing to MAXDOAS data. Values for GDP 4.9 (this study) are given and the values for GDP 4.8 are reported in brackets for comparison. Results for both the original comparisons and the smoothed comparisons (smo.) are reported.

| | AD ± STDEV ($\times 10^{15}$); RD (%) | R | regression parameters |
|---|---|---|---|
| Beijing | -16±7.3; -47% [-21±4.5; -60%] | 0.94 [0.95] | S=0.4±0.05, I=3.4±0.6 [S=0.58±0.06, I=-6.2±0.7] |
| Beijing (smo.) | -11±6.5; -37% [-16±6.3; -52%] | 0.94 [0.96] | S=0.43±0.05, I= 4.4±0.6 [S=0.48±0.04, I= 0.11±0.5] |
| Bujumbura | -3.6±1.8; -76% [-3.7±1.1; -89%] | na [0.29] | na [S=0.1±0.05, I=0.012±0.12] |
| Bujumbura (smo.) | -1.9±1.2; -62% [-2.4±0.8; -84%] | na [0.51] | na [S=0.22±0.06, I=-0.18±0.1] |
| OHP | -0.85±1; -25% [-1.2±0.7; -45%] | 0.4 [0.69] | S=0.25±0.06, I=1.2±0.1 [S=0.73±0.07, I=-0.5±0.1] |
| Reunion | -1.5±0.5; -64% [-1.9±0.4; -90%] | 0.14 [0.23] | S=0.05±0.12, I=0.64±0.2 [S=0.06±0.06, I=0.12±0.08] |
| Reunion (smo.) | -0.4±0.4; -31% [-0.7±0.2; -77%] | 0.15 [0.28] | S=0.12±0.25, I=0.06±0.09 [S=0.32±0.25, I=-0.01±0.2] |
| Uccle | -5±2.7; -43% [-6.2±3.7; -52%] | 0.82 [0.49] | S=0.47±0.04, I=0.83±0.2 [S=0.35±0.08, I=1.1±0.4] |
| Uccle (smo.) | -3.8±2.8; -34% [-7.6±4.3; -57%] | 0.75 [0.51] | S=0.45±0.05, I=0.15±0.05 [S=0.28±0.06, I=1.5±0.3] |
| Xianghe | -2.7±5.3; -5.8% [-9.2±7.1; -30%] | 0.91 [0.86] | S=0.72±0.04, I=4.2±0.5 [S=0.63±0.04, I= 1.3±0.5] |
| Xianghe (smo.) | -6.1±8.8; -13% [-11±9.6; -32%] | 0.92 [0.9] | S=0.52±0.03, I= 7.4±0.4 [S=0.48±0.03, I=4.3±0.5] |

**Table 5.** Same as Table 4 but for GOME-2B product.

| | AD ± STDEV ($\times 10^{15}$); RD (%) | R | regression parameters |
|---|---|---|---|
| Bujumbura | -2.8±0.9; -74% [-3.4±1; -87%] | 0.14 [0.09] | S=0.05±0.06, I=0.76±0.12 [S=0.03±0.06, I=0.34±0.1] |
| Bujumbura (smo.) | -1.3±0.7; -57% [-2±0.8; -81%] | 0.28 [0.35] | S=0.14± 0.06, I=0.06±0.04 [S=0.15±0.06, I=0.08±0.1] |
| OHP | -0.5±0.7; -17% [-1±0.6; -42% ] | 0.13 [0.52] | S=0.11±0.13, I=1.5±0.2 [S=0.82±0.2, I=-0.6±0.3] |
| Reunion | -0.8±0.3; -47% [-1.6±0.3; -86%] | 0.56 [0.26] | S=0.71±0.4, I=-0.36±0.52 [S=0.08±0.06, I=0.13±0.09] |
| Reunion (smo.) | 0.05±0.2; 6.7% [-0.5±0.2; -64%] | 0.78 [0.14] | S=-2.5±0.8, I=-0.12±0.22 [S=0.38±0.6, I=0.004±0.5] |
| Uccle | -4.2±2.4; -40% [-5.6±3.1; -54%] | 0.71 [0.71] | S=0.53±0.09, I=0.47±0.4 [S=0.64±0.1, I=-1.7±0.5] |
| Uccle (smo.) | -2.8±2.5; -29% [-6.8±3.4; -56%] | 0.69 [0.73] | S=0.53±0.09, I=0.13±0.09 [S=0.52±,0.1 I= -1±0.4] |
| Xianghe | -3±9.4; -2.2% [-8.4±8.7; -26%] | 0.87 [0.84] | S=0.49±0.04, I=9.6±0.66 [S=0.6±0.05, I=2.5±0.7] |
| Xianghe (smo.) | -6.4±13; -11% [-11±12; -27%] | 0.89 [0.89] | S=0.38±0.03, I=11±0.6 [S=0.46±0.03, I=5.2±0.58] |

The comparisons of satellite and smoothed MAXDOAS columns for the different stations are reported in the supplement (Fig. S5 and S6) and Table 4 and 5. The different impact of MAXDOAS smoothing on the 2 GDP products results from the different AK as parameters like surface albedo or a priori NO$_2$ profiles used in both satellite retrievals are quite different (see Sect. 6). In general, the use of smoothing reduces the MAXDOAS columns and thus reduces both the daily and monthly differences of satellite and MAXDOAS columns. When the average kernels are used to remove the contribution of a priori NO$_2$

profile shape, as seen in Table 4 and 5 and in Fig. S5 and S6, the relative differences in urban conditions such as in Beijing or Uccle decrease from [-52,-57]% for GDP 4.8 to [-34,-37]% for GDP 4.9 for GOME-2A and from -56% to -29% for GOME-2B. The differences go from -32% to -13% for GOME-2A and from -27% to -11% for GOME-2B for suburban conditions such as in Xianghe and go from -77% to -31% for GOME-2A and from -64% to -7% for GOME-2B for remote conditions such as in Reunion.

The results obtained here are coherent with other validation exercises at different stations and with other satellite products, where the $NO_2$ levels are underestimated by the satellite sensors, e.g., with differences of 5% to 25% over China (Ma et al., 2013; Wu et al., 2013; Wang et al., 2017; Drosoglou et al., 2018), mostly explained by the relatively low sensitivity of space-borne measurements near the surface, the gradient-smoothing effect, and the aerosol shielding effect. These effects are often inherent to the different measurements types or the specific conditions of the validation sites (as seen for the different results for Beijing and Xianghe sites in this manuscript), but also to the remaining impact of structural uncertainties (Boersma et al., 2016), such as the impact of the choices of the a priori $NO_2$ profiles and/or the albedo database assumed for the satellite AMF calculations (see Sect. 6). Lorente et al. (2017) estimated e.g. the AMF structural uncertainty to be on average 42% over polluted regions and 31% over unpolluted regions, mostly driven by substantial differences in the a priori trace gas profiles, surface albedo and cloud parameters used to represent the state of the atmosphere. However, the differences in Bujumbura are still of -62%, because of the peculiar condition with the MAXDOAS being in a valley, close to the Tanganika lake, which always leads to a higher surface pressure for the satellite pixels due to the information coming from the a priori model. This is leading to large representation errors and uncertainties in the comparisons (Boersma et al., 2016) that needs to be investigated in more details.

## 8  Conclusions

$NO_2$ columns retrieved from measurements of the GOME-2 aboard the MetOp-A and MetOp-B platforms have been successfully applied in many studies. The abundance of $NO_2$ is retrieved from the narrow band absorption structures of $NO_2$ in the backscattered and reflected radiation in the visible spectral region. The current operational retrieval algorithm (GDP 4.8) for total and tropospheric $NO_2$ from GOME-2 was first introduced by Valks et al. (2011), and an improved algorithm (GDP 4.9) is described in this paper.

To calculate the $NO_2$ slant columns, a larger 425-497 nm wavelength fitting window is used in the DOAS fit to increase the signal-to-noise ratio. Absorption cross-sections are updated and a linear intensity offset correction is applied. The long-term and in-orbit variations of GOME-2 slit function are corrected by deriving effective slit functions with a stretched preflight GOME-2 slit function and by including a resolution correction function as a pseudo absorber cross-section in the DOAS fit, respectively. Compared to the GDP 4.8 algorithm, the $NO_2$ columns from GDP 4.9 are higher by $\sim$1-3 $\times 10^{14}$ molec/cm$^2$ (up to 27%) and the $NO_2$ slant column noise is lower by $\sim$24%. In addition, the effect of using a new version (6.1) of the GOME-2 level 1b data has been analyzed in our $NO_2$ algorithm. The application of new GOME-2 level 1b data largely reduces the offset between GOME-2A and GOME-2B $NO_2$ columns by removing calibration artefacts in the GOME-2B irradiances (due

to Xe-line contaminations in the calibration key-data). Compared to the GOME-2 NO$_2$ product from the QA4ECV project, the NO$_2$ columns from GDP 4.9 show good consistency and the NO$_2$ slant column noise is $\sim$14%-28% smaller, indicating a good overall quality of the improved DOAS retrieval.

The stratosphere-troposphere separation algorithm STREAM, which was designed for TROPOMI, was optimized for GOME-2 instrument. Compared to the spatial filtering method used in the GDP 4.8, STREAM provides an improved treatment of polluted and cloudy pixels by defining weighting factors for each measurement depending on polluted situation and cloudy information. For the adaption to GOME-2 measurements, the performance of STREAM is analyzed by applying it to synthetic GOME-2 data and by comparing the difference between estimated and original stratospheric fields. Applied to synthetic GOME-2 data calculated by a RTM using C-IFS model data, the estimated stratospheric NO$_2$ columns from STREAM show good consistency with the a priori truth. A slight overestimation by $\sim$1-2 $\times 10^{14}$ molec/cm$^2$ is found over lower latitudes, and larger differences of up to $\sim$5 $\times 10^{14}$ molec/cm$^2$ are found at higher latitudes. To reduce the biases over the subtropical regions in winter, an improved latitudinal correction is used in STREAM. Applied to GOME-2 measurements, the updated STREAM separates successfully the stratospheric and tropospheric contribution over polluted regions, especially in the northern hemisphere. Compared to the current method in the GDP 4.8, the use of STREAM slightly decreases the stratospheric NO$_2$ columns by $\sim$1 $\times 10^{14}$ molec/cm$^2$ in general and largely reduces the overestimation over polluted areas.

To improve the calculation of NO$_2$ AMF, a new box-AMF LUT was generated using the latest version of the VLIDORT RTM with an increased number of reference points and vertical layers to reduce interpolation errors. The new GOME-2 surface LER climatology (Tilstra et al., 2017b) used in this study is derived with a high resolution of 1$°$lon$\times$1$°$lat (0.25$°$lon$\times$0.25$°$lat at coastlines) and an improved LER algorithm based on observations for 2007-2013. Daily a priori NO$_2$ profiles, obtained from the chemistry transport model TM5-MP, capture the short-term variability in the NO$_2$ fields with a resolution of 1$°$lon$\times$1$°$lat. A large impact on the retrieved tropospheric NO$_2$ columns (more than 10%) is found over polluted areas.

The uncertainty in our GDP 4.9 NO$_2$ slant columns is $4.4 \times 10^{14}$ molec/cm$^2$, calculated from the average slant column error using a statistical method described in Sect. 4.5. The uncertainty in the GOME-2 stratospheric columns is $\sim$4-5 $\times 10^{14}$ molec/cm$^2$ for polluted conditions based on the daily synthetic GOME-2 data and $\sim$1-2$\times 10^{14}$ molec/cm$^2$ for monthly averages. The uncertainty in the tropospheric AMFs is estimated to be in the 10-45% range, considering the use of updated box-AMF LUT and improved surface albedo climatology and a priori NO$_2$ profiles, resulting in a total uncertainty in the tropospheric NO$_2$ columns likely in the range of 30-70% for polluted conditions.

An end-to-end validation of the improved GOME-2 GDP 4.9 dataset was performed by comparing the GOME-2 tropospheric NO$_2$ columns with BIRA-IASB ground-based MAXDOAS measurements. The validation was illustrated for different MAXDOAS stations (Beijing, Bujumbura, OHP, Reunion, Uccle, and Xianghe) covering urban, suburban, and background situations. Taking Xianghe station as an example, the GDP 4.9 dataset shows a similar seasonal variation in the tropospheric NO$_2$ columns as the MAXDOAS measurements with a relative difference of -5.8% (i.e. $-2.7 \times 10^{15}$ molec/cm$^2$ in absolute) and a correlation coefficient of 0.91 for GOME-2A, indicating a good agreement. The Xianghe site, by its suburban nature, is the best site for validation. At the other sites, mean biases ranges from [-47%; -16 $\times 10^{15}$ molec/cm$^2$] for Beijing, [-76%,-74%; -3.6 $\times 10^{15}$ molec/cm$^2$,-2.8 $\times 10^{15}$ molec/cm$^2$] for Bujumbura, [-25%,-17%; -0.9 $\times 10^{15}$ molec/cm$^2$,-0.5 $\times 10^{15}$ molec/cm$^2$]

for OHP, [-64%,-47%; $-1.5 \times 10^{15}$ molec/cm$^2$,$-0.8 \times 10^{15}$ molec/cm$^2$] for Reunion, and [-43%,-40%; $-5 \times 10^{15}$ molec/cm$^2$,-$4.2 \times 10^{15}$ molec/cm$^2$] for Uccle. Reunion and Bujumbura are difficult sites for validation, due to their valley/mountain nature, while urban sites Beijing and Uccle show similar relative results. The smaller absolute bias is found at the rural OHP station. Compared to the current operational GDP 4.8 product, the GDP 4.9 dataset is a significant improvement. Although GOME-2 measurements are still underestimating the tropospheric NO$_2$ columns with respect to the ground data, the absolute and relative differences with the different MAXDOAS stations are smaller, both for the original comparisons and for the comparisons with the smoothed MAXDOAS columns.

In the future, the AMF calculation will be further improved, since uncertainty in AMF is one dominating source of errors in the tropospheric NO$_2$ retrieval, especially over polluted areas. The surface Bidirectional Reflectance Distribution Function (BRDF) effect will be included using a direction-dependent LER climatology from GOME-2 (Tilstra, L., personal communication) to describe the angular distribution of the surface reflectance. Aerosol properties will be considered explicitly in the RTM calculation using ground-based aerosol observations from e.g. MAXDOAS instruments, Mie scattering Lidars, or sun photometers operated by the AErosol RObotic NETwork (AERONET). A priori NO$_2$ profiles from different global and regional models will help to analyse the effect of spatial resolution, temporal resolution, and emission on the tropospheric NO$_2$ retrieval for GOME-2. Furthermore, the NO$_2$ algorithm will be adapted to measurements from the TROPOMI instrument with a spatial resolution as high as $7 \times 3.5$ km$^2$.

*Acknowledgements.* This work is funded by the DLR-DAAD Research Fellowships 2015 (57186656) programme with reference number 91585186 and is undertaken in the framework of the EUMETSAT AC-SAF project. We acknowledge the Belgian Science Policy Office (BELSPO) supporting part of this work through the PRODEX project B-ACSAF. We thank EUMETSAT for the ground segment interfacing work and for the provision of GOME-2 level 1 products. We thank the UPAS team for the development work on the UPAS system at DLR. We are thankful to Rüdiger Lang (EUMETSAT) for providing the GOME-2 level 1b testing data, Vincent Huijnen (KNMI) for providing the C-IFS model data, Gijsbert Tilstra (KNMI) for discussions on surface albedo, and Henk Eskes (KNMI) for creating the TM5-MP a priori NO$_2$ profiles. We acknowledge the free use of GOME-2 NO$_2$ column data from QA4ECV project available at www.qa4ecv.eu. We also acknowledge the free use of the GOME-2 surface LER database created by KNMI and provided through the AC-SAF of EUMETSAT.

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
