# Peer review of "An Improved Total and Tropospheric NO2 Column Retrieval for GOME-2"

_Atmospheric Measurement Techniques, 2018_

## Referee Comment (RC1) · K. Yang (Referee) · 8 Sep 2018

This paper describes the many algorithm improvements implemented in a future version (4.9) of GDP used by the EUMETSAT AC-SAF for NO2 retrieval from GOME-2. Among them, the main improvements are a broader fitting window, 425 – 497 nm (extended from GDP4.8, 425 – 450 nm) and the associated linear intensity offset correction, and the adaptation of STREAM for its stratosphere-troposphere NO2 separation scheme. Though most (if not all) these improvements have been presented previously in various published works, this paper documents these changes coherently. Additionally, this paper includes tropospheric NO2 comparison with ground-based MAXDOAS measurements, in an effort to validate the improved algorithm. This paper may be viewed as the algorithm document for GDP4.9, which will be applied to the series of

GOME-2 instruments for generating a long-term NO2 record. It is suitable for publication in AMT. There are some clarifications and minor edits needed for acceptance.

Specific Comments:

1. Validation

The validation using MAXDOAS suggests that GPD4.9 agrees better with this independent correlative measurements than GDP4.8, because the bias reductions (shown in Tables 4 and 5) achieved with the newer version GDP. But this overemphasizes the bias reduction. The equally important measure is the standard deviation, which shows little or no improvement. In fact, there are many cases (Tables 4 and 5) that show large standard deviation or lower correlation with MAXDOAS for the new GDP, indicating that the agreement becomes worse with the algorithm changes. One could argue that biases in satellite measurements may be easier to remove, often achieved by offset adjustments. Therefore lower biases likely do not say much about retrieval improvements. Keeping in mind that the coincidence of the ground-based MAXDOAS and GOME-2 may be limited, the agreements (as measured by standard deviation of difference or correlation coefficient) have its limitation as well. I recommend revise section 7, and add some discussions on the agreement based on standard deviation and correlation.

2. Larger Fitting Window

While it is certainly true that the noise level of slant NO2 column is lower for DOAS retrieval from a larger fitting window in general, there is a downside as well. The key assumption of DOAS approach is that the AMF is (nearly) independent of wavelength. However as the fitting window becomes larger, the spectral dependence of tropospheric AMF becomes more prominent. For instance, the measurement sensitivity (called the box AMF ml, Eq.8) is ∼20% higher at 490 nm than at 425 nm at 1 km above the surface, for a spectral invariant surface reflectivity at 0.1. Closer to the surface at 0.1 km, ml is ∼30% higher at 490 than at 425 nm. Furthermore, surface reflectivity depends on wavelength as well, introducing additional spectral variation in

the tropospheric AMF. This spectral variation in AMF implies that the absorption signal would be 20% to 30% stronger at 490 nm than at 425 nm for the same amount tropospheric NO2. Considering that many improvements described in the paper are on the order of a few percent, perhaps it is a good idea to discuss how the AMF spectral variation affects the retrieval accuracy. My rough estimate indicates that it may have up to 15% error for the larger fitting window when neglecting the spectral variation.

3. Intensity Offset

* Based on Eq. 1, the offset listed in table 1, should simply be 'a' for GDP4.8, and 'a + b $\lambda$' for GDP4.9, not ln(I+a) and ln(I+a+b $\lambda$), respectively.

* Please add in the paper a description of how the parameters 'a' and 'b' are determined.

* The additive intensity offset looks similar to stray light contribution. It would be helpful add some plots to accompany the difference plots in Fig. 1. Specifically, include a map of LER to show the scene reflectivity, and maps of the parameters 'a' and 'b'. If the offset has something to with stray lights, 'a' would be high for low reflectivity scene, and vice versa, assuming that L1B data are not yet stray-light corrected. In any case, these additional plots would reveal information about L1B, and lend credibility of the intensity-offset correction.

4. Minor edits

* Page 2, line 9: "a strong growth of NO2 since two decades has caused", 'since two decades' is not clear, revise please.

* Page 2, line 33: "measured GOME-2 (ir)radiances", should be measured GOME-2 radiances, or measured GOME-2 sun-normalized radiances.

* Page 22, line 10: "and photon path", not specific or meaningful here. May be removed.

---

## Referee Comment (RC2) · Anonymous Referee #2 · 13 Sep 2018

This study presents an improved retrieval of SCD and VCD of NO2 from Global Ozone Monitoring Experiment-2 (GOME-2), based on an updated GDP 4.9 algorithm. The topic of the manuscript is within the scope of AMT and it is of interest to the scientific community. Below are a few comments.

Major comments

While the paper presents a good description of SCD calculation and stratosphere-troposphere separation, its discussion on tropospheric AMF and VCD is relatively weak. 1) The paper includes updates on surface albedo (from TOMS-based to GOME2-based) and NO2 profile (from monthly climatology to daily varying). The importance of these updates are well known (e.g., see OMNO2, DOMINO, QA4ECV and POMINO for OMI), thus the significance of the finding here should be presented in a

way that it confirms (or contrasts against) the findings of previous studies, aided with sufficient relevant citations in Sect. 1 and 6.3. Also, since the surface albedo climatology is used, what would be the implication of ignoring interannual variability and trends of albedo (as shown in many land cover change studies) on the AMF? 2) For tropospheric AMF, the most important sources of errors come from cloud retrievals and aerosols (e.g., Lorente et al., 2017 and references therein). However, the description of cloud retrieval and (especially) how it is affected by other ancillary parameters (e.g., from TOMS to GOME2 surface albedo) is not clear (e.g., in Sect. 6.4). 3) The crucial role of aerosol treatment (in polluted cases) is only vaguely described, and should thus be better analyzed. If explicitly representing aerosols in the algorithm is not possible, a better description of implied errors should be presented, aided with relevant citations. 4) A global map of tropospheric NO2 VCD (e.g., monthly climatology, not just one specific month) and how these algorithm updates/uncertainties affect the VCD should be given and discussed.

At its current form, the comparison with MAX-DOAS data does not tell much information regarding the causes of satellite errors (especially slope and bias), thus providing little new knowledge beyond previous findings. The discussion on sampling difference and retrieval algorithm should be aided with quantitative tests (of the algorithm assumptions/parameters, within the context of this study), particularly on how the data sampling (50 km is a relatively loose criterion; the temporal matching is not clear), cloud treatment and aerosol treatment each affects the comparison. Also, the main text only emphasizes the comparison at Xianghe (Figs. 14-16), which gives best consistency. However, the SI shows much poorer comparisons (in terms of bias and slope) at other sites. A better description in the main text of these comparisons (e.g., by better reference to SI, or moving some figures from SI to the main text) is needed to accurately present the overall quality of the satellite product.

Specific comments:

Title: change "total" to "slant". The term "total column" is misleading.

P1, L18: Please give the full name of VLIDORT and clarify which version

P1, L20: "a large effect" is somehow ambiguous. Does it mean decrease or increase?

P2,L6 – VOC is not necessary for aerosol formation from NOx.

The title and content of Sect. 3 should be clarified such that the reader understands the section is describing the old algorithm.

P3,L16-17 – the treatment of aerosol is crucial especially for polluted cases.

Table 1 – check the formulas of intensity offset for QA4ECV

Figure 2, right panel – discuss the discontinuity in 2015

Figure 6 – the error appears to be stabilized after 2010. Please discuss.

P15,L10 – "latitudinal" should be "longitudinal"

P19, the paragraph "Figure 11 illustrates. . ." – what is the impact on cloud retrieval, and how would this affect the sampling (with criterion of CRF < 0.5)?

P22, L8: All discussion in this section only shows the improvement of cloud retrieval itself. How about its influence on NO2 retrieval?

P25,L3 – test the choice of 50 km, which is relatively loose. Also, how is the temporal interpolation done?

P26, first paragraph – need a better description of comparisons at other sites (see my major comment)

P26,L8-11 – The explanation is qualitative. A more quantitative explanation of slope and bias is needed. For example, to what extent the slope and bias can be explained by sampling difference and algorithm limitations? See my major comment.

P28,L9-10 – the sentence is not clear.

P28,L17-21 – again, the statement is too general.

P29,L32 – the uncertainty in tropospheric NO2 AMF appears underestimated, especially given the large error in cloud and aerosol treatment (e.g., Lorente et al., 2017).

P30, first paragraph – the consistency at other sites is much poorer. This point should be presented here.
* * *

---

## Author Comment (AC1) · 24 Nov 2018

We thank the reviewer for the careful reading and the critical thinking, especially about the wavelength-dependency in AMF calculation and the MAXDOAS validation. Our response to the comments and detailed changes made to the manuscript are described below in black and blue, respectively, including the reviewer's text in red.

Major comments

1. Validation
The validation using MAXDOAS suggests that GPD4.9 agrees better with this independent correlative measurements than GDP4.8, because the bias reductions (shown in Tables 4 and 5) achieved with the newer version GDP. But this overemphasizes the bias reduction. The equally important measure is the standard deviation, which shows little or no improvement. In fact, there are many cases (Tables 4 and 5) that show large standard deviation or lower correlation with MAXDOAS for the new GDP, indicating that the agreement becomes worse with the algorithm changes. One could argue that biases in satellite measurements may be easier to remove, often achieved by offset adjustments. Therefore lower biases likely do not say much about retrieval improvements. Keeping in mind that the coincidence of the ground-based MAXDOAS and GOME-2 may be limited, the agreements (as measured by standard deviation of difference or correlation coefficient) have its limitation as well. I recommend revise section 7, and add some discussions on the agreement based on standard deviation and correlation.

We agree with the need of emphasizing the standard deviation and correlation, leading to the revision of Sect. 7 page 26-28 with more discussions. As a response to the second reviewer, we have also added more discussions in terms of slope and bias as well as more descriptions at other stations besides Xianghe (which gives the best validation results). Please refer to the second response letter page 6.

2. Larger Fitting Window
While it is true that the noise level of slant $NO_2$ column is lower for DOAS retrieval from a larger fitting window in general, there is a downside as well. The key assumption of DOAS approach is that the AMF is (nearly) independent of wavelength. However as the fitting window becomes larger, the spectral dependence of tropospheric AMF becomes more prominent. For instance, the measurement sensitivity (called the box AMF ml, Eq.8) is 20% higher at 490 nm than at 425 nm at 1 km above the surface, for a spectral invariant surface reflectivity at 0.1. Closer to the surface at 0.1 km, ml is 30% higher at 490 than at 425 nm. Furthermore, surface reflectivity depends on wavelength as well, introducing additional spectral variation the tropospheric AMF. This spectral variation in AMF implies that the absorption signal would be 20% to 30% stronger at 490 nm than at 425 nm for the same amount tropospheric $NO_2$. Considering that many improvements described in the paper are

on the order of a few percent, perhaps it is a good idea to discuss how the AMF spectral variation affects the retrieval accuracy. My rough estimate indicates that it may have up to 15% error for the larger fitting window when neglecting the spectral variation.

[Figure]

ALB=0.1 SZA=60° VZA=40° RAA=135°

Figure I: The retrieved tropospheric $NO_2$ columns as a function of wavelength for a polluted boundary layer with $1 \times 10^{16}$ molec/cm$^2$ (horizontal line), 1 km boundary layer height, surface albedo of 0.1, SZA of 60°, VZA of 40°, RAA of 135°, and without aerosol.

The different fitting window results in different sensitivities to boundary layer pollution, especially when the surface albedo varies with wavelength at the same time. To evaluate the error introduced by the wavelength-dependency of AMF to our $NO_2$ retrieval, we have implemented a sensitivity study using the LIDORT radiative transfer model. We simulated clear-sky TOA reflectances (and also tropospheric AMFs) between 400-500 nm with a 0.22 nm step for a typical polluted scenario with a "true" tropospheric $NO_2$ column $1 \times 10^{16}$ molec/cm$^2$. Given the slant column of $2.2 \times 10^{16}$ molec/cm$^2$ that was fitted using the 425-497 nm fitting window, the retrieved tropospheric $NO_2$ column as a function of wavelength (using the wavelength-dependent tropospheric AMFs) is shown in Fig. I. The tropospheric $NO_2$ column decreases (i.e. the tropospheric AMF increases) between 425-497 nm by 30.9% due to the stronger Rayleigh scattering, similar to the values suggested by the reviewer. However, the retrieval accuracy may not be affected that much, since the tropospheric $NO_2$ column calculated at the mid-point wavelength of fitting window (461 nm) is close to the true value (with a difference of 3.7%). Similar results are also obtained for other geometries and surface albedo. Since the discussion for surface albedo has been included in Sect. 6.2 with an impact on the tropospheric $NO_2$ column by up to 5%, we have extended the discussion for AMF calculation in page 18 line 10:

The box-AMFs $m_l$ for each layer are calculated for the mid-point wavelength of fitting window, i.e., 461 nm in our $NO_2$ retrieval, which is representative of the window-average box-AMFs. Compared to the tropospheric AMFs at 440 nm (mid-point wavelength in GDP 4.8), the ones calculated at 461 nm are higher by up to 10% for polluted situations, due to the wavelength-dependency of Rayleigh scattering, in agreement with Boersma et al. (2018) (see Fig 7 therein). Note that the uncertainty related to the wavelength-dependency of the AMF is much smaller than the uncertainties introduced by surface albedo, a priori $NO_2$ profile, cloud and aerosol (see Sect. 6.4).

3. Intensity Offset
* Based on Eq. 1, the offset listed in table 1, should simply be 'a' for GDP4.8, and 'a + bλ' for GDP4.9, not ln(I+a) and ln(I+a+bλ), respectively.

To make the comparison less confusing, we have removed the formulas from the table and classified the intensity offset simply as "constant" vs. "linear" in wavelength.

* Please add in the paper a description of how the parameters 'a' and 'b' are determined.

The offset correction is modelled using a polynomial and fitted as a non-linear parameter in QDOAS. We have added the method for GDP 4.8 in page 4 line 20:

The intensity offset, which describes the additional contributions such as stray light in the spectrometer to the measured intensity, is modelled using a zero order polynomial with polynomial coefficient as fitting parameter.

We have added the updates for GDP 4.9 in page 7 line 27:

To correct for this drift, an intensity offset correction with a linear wavelength dependency (i.e., polynomial degree of 1) is applied for the large fitting window in this study.

The additive intensity offset looks similar to stray light contribution. It would be helpful add some plots to accompany the difference plots in Fig. 1. Specifically, include a map of LER to show the scene reflectivity, and maps of the parameters 'a' and 'b'. If the offset has something to with stray lights, 'a' would be high for low reflectivity scene, and vice versa, assuming that L1B data are not yet stray-light

corrected. In any case, these additional plots would reveal information about L1B, and lend credibility of the intensity-offset correction.

Figure II shows the parameter 'a' on 05 Feb 2009 (instead of the given day in March in the manuscript so that the LER map in Fig. 10 can be referable). The intensity offset depends on the stray light in the instrument, but the dependency of 'a' (and also 'b', not shown) on LER seems not dominant. Given the combined impact of several factors on the intensity offset, this is not surprising. The dominating structures in 'a' are related to the ocean-land contrast (vibrational raman scattering), the orbital swath (instrument calibration issue related to the scanning mirror), and the cloud pattern (e.g., cloud albedo or fraction, possibly related to the stray light or incomplete removal of Ring effect). Since a stray light correction has been included during the GOME-2 level 0 to 1b processing, the residual stray light might have a relatively small impact, compared to the factors described above.

[Figure]

Figure II: The fitted (constant) polynomial coefficient for intensity offset correction (top left), the cloud albedo (top right), difference in $NO_2$ columns (slant columns scaled by geometric AMFs) (bottom left) and retrieval RMS (bottom right) estimated with and without intensity offset correction for GOME-2A on 5 February 2009.

4. Minor edits

* Page 2, line 9: "a strong growth of $NO_2$ since two decades has caused", 'since two decades' is not clear, revise please.

We have changed to:

a strong growth of $NO_2$ during the past two decades has caused

* Page 2, line 33: "measured GOME-2 (ir)radiances", should be measured GOME-2 radiances, or measured GOME-2 sun-normalized radiances.

We have changed to:

GOME-2 sun-normalized radiances

* Page 22, line 10: "and photon path", not specific or meaningful here. May be removed.

Done.

**References**

Boersma, K., Eskes, H., Richter, A., De Smedt, I., Lorente, A., Beirle, S., Van Geffen, J., Zara, M., Peters, E., Van Roozendael, M., Wagner, T., Maasakkers, J., van der A, R., Nighttingale, J., De Rudder, A., Irie, H., Pinardi, G., Lambert, J.-C., and Compernolle: Improving algorithms and uncertainty estimates for satellite NO2 retrievals: Results from the Quality Assurance for Essential Climate Variables (QA4ECV) project, submitted, 2018.

Müller, J.-P., Kharbouche, S., Gobron, N., Scanlon, T., Govaerts, Y., Danne, O., Schultz, J., Lattanzio, A., Peters, E., De Smedt, I., Beirle, S., Lorente, A., Coheur, P., George, M., Wagner, T., Hilboll, A., Richter, A., Van Roozendael, M., and Boersma, K.: Recommendations (scientific) on best practices for retrievals for Land and Atmosphere ECVs, Tech. rep., Deliverable 4.2 - version 1.0, 2016.

Richter, A. and verification team, S.-P.: S5P/TROPOMI Science Verification Report, Tech. rep., S5P-IUP-L2-ScVR-RP issue 2.1, 2015.

---

## Author Comment (AC2) · 24 Nov 2018

We thank the reviewer for the careful reading and the critical thinking, especially about the tropospheric AMF calculation and the MAXDOAS validation. Our response to the comments and detailed changes made to the manuscript are described below in black and blue, respectively, including the reviewer's text in red.

Major comments

1) The paper includes updates on surface albedo (from TOMS-based to GOME2-based) and $NO_2$ profile (from monthly climatology to daily varying). The importance of these updates are well known (e.g., see OMNO2, DOMINO, QA4ECV and POMINO for OMI), thus the significance of the finding here should be presented in a way that it confirms (or contrasts against) the findings of previous studies, aided with sufficient relevant citations in Sect. 1 and 6.3.

We have added more references in Sect. 1 to recent studies on surface albedo, $NO_2$ profiles, cloud and aerosol for OMI in page 3 line 16:

The AMFs are determined with a radiative transfer model (RTM) and stored in a look-up table (LUT) requiring ancillary information such as surface albedo, vertical shape of the a priori $NO_2$ profile, clouds and aerosols. Improvements in the RTM and LUT interpolation scheme, the ancillary parameters, and the cloud and aerosol correction approach have been reported for OMI instrument (e.g., Boersma et al., 2011; Lorente et al., 2017; Vasilkov et al., 2017; Krotkov et al., 2017; Veefkind et al., 2016; Lin et al., 2014; Castellanos et al., 2015; Laughner et al., 2018), which in principle are beneficial for similar satellite instruments like GOME-2.

Most of the above surface albedo studies use the MODIS BRDF product to improve the AMF calculation. This can however introduce additional biases due to the use of different instruments. Therefore, the significance of the finding here is related to the use of the latest improved version of the GOME-2 LER climatology that is consistent with the $NO_2$ retrieval from the same instrument, as introduced in Sect. 6.2.

We have added more references in Sect. 6.3 focusing on the updates of the a priori $NO_2$ profile, mainly the use of higher spatial resolution profiles and the use of daily profiles instead of monthly profiles, in page 20 line 9:

Increasing the spatial and/or temporal resolution of the a priori profiles have shown to produce a more accurate $NO_2$ retrieval (e.g., Russell et al., 2011; Heckel et al., 2011; McLinden et al., 2014; Nüß et al., 2006; Laughner et al., 2016).

We have related our findings about the use of daily vs. monthly profiles, which partly explains the large difference between the IMAGES used in GDP 4.8 and the

TM5-MP in GDP 4.9, to previous works in page 22 line 3:

In agreement with Nüß et al. (2006) and Laughner et al. (2016), the use of monthly profiles changes the tropospheric $NO_2$ columns by up to $3 \times 10^{15}$ molec/cm$^2$ depending on the wind speed and wind direction, in particular for regions affected by transport (not shown).

Also, since the surface albedo climatology is used, what would be the implication of ignoring interannual variability and trends of albedo (as shown in many land cover change studies) on the AMF?

As analyzed by Sütterlin et al. (2016) for Europe using the AVHRR/BRDF albedo product for the year 1990-2014, the land surface albedo (in the visible wavelength) decreases by $\sim$0.004 per decade, probably driven by the change of land cover, vegetation, snow or ice. The interannual variability of the land surface albedo is generally smaller than 0.01 for snow-free vegetation cover but possibly larger than 0.06 for regions affected by snow. For our $NO_2$ retrieval, ignoring the decreasing trend of surface albedo leaves a small impact on the tropospheric $NO_2$ columns by up to $2 \times 10^{14}$ molec/cm$^2$ (absolute) and 3% (relative). However, ignoring the interannual variability can introduce large errors in the AMF calculation for the varying snow and ice situation. Possible corrections for the surface albedo climatology include the use of external information about the actual snow and ice conditions, e.g., from Near-real-time Ice and Snow Extent (NISE) dataset. It is worth noting that reducing the error introduced by the interannual variability and trends was also a main motivation of using more recent observations for the LER climatology from GOME-2.

2) For tropospheric AMF, the most important sources of errors come from cloud retrievals and aerosols (e.g., Lorente et al., 2017 and references therein). However, the description of cloud retrieval and (especially) how it is affected by other ancillary parameters (e.g., from TOMS to GOME2 surface albedo) is not clear (e.g., in Sect. 6.4).

Lorente et al. (2017) estimated that the tropospheric AMF error is substantial related to the cloud correction approach (IPA vs. clear-sky AMF) by 5-40%, the aerosol correction approach (implicit vs. explicit) by $\sim$50%, and the choice of surface albedo, surface height, a priori $NO_2$ profiles, and cloud parameters. Boersma et al. (2018) concluded that the largest cloud parameters-related uncertainty is introduced by the surface albedo-cloud fraction error correlation by 10-20% using OMCLDO2 cloud parameters. For the OCRA/ROCINN cloud algorithm, the surface albedo-cloud fraction error correlation is relatively small. Unlike cloud algorithm that retrieves the cloud parameters from the TOA reflectance in the $O_2$-$O_2$ absorption band (OMCLDO2) or $O_2$-A absorption band (FRESCO), OCRA calculates the cloud fraction by separating a spectral scene into cloudy contribution and cloud-free background

using spectral information from the UV-VIS-NIR band. The surface albedo is not directly needed as an input in OCRA's cloud fraction retrieval, but it will affect the measured clear-sky TOA reflectances and thus the cloud-free map construction, which mainly influences the cases with rapidly varying surface conditions (e.g. fresh snow or melting) that happen on faster timescales than the monthly resolution of the clear-sky maps. In order to minimize this effect, OCRA linearly interpolates between two monthly maps to "daily" values. Therefore, the surface albedo leaves a relatively smaller impact on our cloud fraction retrieval, compared to other cloud algorithms like OMCLDO2 or FRESCO that require a surface albedo climatology. We have added the discussion about the cloud-related uncertainty in a new section describing the combined effect of algorithm changes and potential uncertainties (the new section is added as a response to comment 4):

The largest cloud-related uncertainty in $NO_2$ retrieval is introduced by the surface albedo-cloud fraction error correlation, as analysed by Boersma et al. (2018) for OMI using OMCLDO2 cloud product, which requires a surface albedo climatology as input in the cloud fraction retrieval. But this uncertainty is likely smaller for OCRA/ROCINN cloud algorithms, since the surface albedo is treated differently in OCRA's cloud fraction calculation. Retrieved by separating a spectral scene into cloudy contribution and cloud-free background, the cloud fraction from OCRA is affected by surface albedo through the cloud-free map construction with a larger impact over bright surfaces like snow or ice cover, in particular during snowfall (higher background) or melting (lower background), which has been corrected by interpolating towards a daily value between two monthly cloud-free map in OCRA (Lutz et al., 2016).

Additionally, we have removed Sect. 6.4 and moved the description of cloud parameters to the general introduction of GDP 4.8 in Sect. 3, since the cloud products are not changed between GDP 4.8 and 4.9.

3) The crucial role of aerosol treatment (in polluted cases) is only vaguely described, and should thus be better analyzed. If explicitly representing aerosols in the algorithm is not possible, a better description of implied errors should be presented, aided with relevant citations.

We have added more references and discussions about aerosol correction in the new section introducing the potential uncertainties (the new section is added as a response to comment 4):

The uncertainty introduced by aerosol in GDP 4.9 is ~50% for high aerosol loading, in agreement with Lorente et al. (2017). With direct impact on $NO_2$ AMF calculation and indirect impact via cloud parameters retrieval, the aerosol effect has been considered for OMI implicitly through the cloud correction (Boersma et al.,

2004, 2011) or explicitly with additional aerosol information for regional studies (Lin et al., 2014, 2015; Kuhlmann et al., 2015; Castellanos et al., 2015; Chimot et al., 2018), leading to an increase or decrease of $NO_2$ AMF by up to 40% depending on $NO_2$ distribution and aerosol properties and distribution. Since aerosol is highly variable in space and time due to the dependency on emission sources, transports, and atmospheric processes (Holben et al., 1991), explicit aerosol correction will be applied in our AMF calculation when reliable observations or model outputs of aerosol optical properties and vertical distributions are available.

We agree with the importance of aerosol treatment in AMF calculation, and we will investigate the explicit correction for the aerosol effect in a following paper, with a case study over China using ground-based measurements from AERONET, Lidar, and MAXDOAS.

4) A global map of tropospheric $NO_2$ VCD (e.g., monthly climatology, not just one specific month) and how these algorithm updates/uncertainties affect the VCD should be given and discussed.

We have added an additional section named "6.4 Examples of GOME-2 tropospheric $NO_2$" to address the combined algorithm changes and uncertainties, with monthly climatology for Feb. and Aug. as examples:

Figure I shows the tropospheric $NO_2$ columns from the improved GDP 4.9 algorithm for February and August averaged for the year 2007-2016. Figure II shows the difference in tropospheric $NO_2$ columns from the GDP 4.9 and GDP 4.8 product. The tropospheric $NO_2$ columns increase globally by $\sim 1 \times 10^{14}$ molec/cm$^2$ due to the improved DOAS slant column fitting and increase further by $\sim 3 \times 10^{14}$ molec/cm$^2$ around moderately polluted regions beneficial from the use of new stratosphere-troposphere separation algorithm STREAM. A stronger change by more than $1 \times 10^{15}$ molec/cm$^2$ is found mainly over polluted continents, as a result of the improvements to the AMF calculation, primarily the surface albedo (which also affects the snow or ice area, e.g., southern Canada and northeastern Europe) and/or the a priori $NO_2$ profiles (which also affects the polluted ocean, e.g., shipping lanes in southeastern Asia).

Over central northern Europe, the tropospheric $NO_2$ columns are reduced by $\sim 1 \times 10^{15}$ molec/cm$^2$ for GDP 4.9 in winter and $\sim 3 \times 10^{14}$ molec/cm$^2$ in summer. A larger number of negative values in GDP 4.8, possibly related to the overestimated stratospheric $NO_2$ around polar vortex areas, is largely corrected in GDP 4.9 by improving the stratosphere-troposphere separation algorithm. Over eastern China and eastern US, the seasonal variation is consistent between GDP 4.8 and 4.9, with reduced values in winter (by more than $1 \times 10^{15}$ molec/cm$^2$) and enlarged values in summer (by more than $1 \times 10^{15}$ molec/cm$^2$ for eastern China and $5 \times 10^{14}$ molec/cm$^2$ for eastern US) for GDP 4.9 due to the combined impact of the algorithm changes, mainly the AMF calculation. Over India and its surrounding areas, a systematic

[Figure]

Figure I: Monthly average tropospheric $NO_2$ columns from GDP 4.9 for clear-sky conditions (cloud radiance fraction smaller than 0.5), measured by GOME-2A in February (left) and August (right) 2007-2016.

[Figure]

Figure II: Difference in tropospheric $NO_2$ columns from GDP 4.9 and GDP 4.8 for clear-sky conditions (cloud radiance fraction smaller than 0.5) in February (left) and August (right) 2007-2016 for GOME-2A.

increase in tropospheric $NO_2$ columns by $\sim 7 \times 10^{14}$ molec/cm$^2$ for GDP 4.9 benefits from the use of STREAM.

The uncertainty in our GDP 4.9 $NO_2$ slant columns is $4.4 \times 10^{14}$ molec/cm$^2$, calculated from the average slant column error using a statistical method described in Sect. 4.5. The uncertainty in the GOME-2 stratospheric columns is $\sim 4-5 \times 10^{14}$ molec/cm$^2$ for polluted conditions based on the daily synthetic GOME-2 data and $\sim 1-2 \times 10^{14}$ molec/cm$^2$ for monthly averages. The uncertainty in the GDP 4.9 AMF calculation is likely reduced, considering the improved surface albedo climatology and a priori $NO_2$ profiles, which are the main causes of AMF structural uncertainty (Lorente et al., 2017). In addition, the AMF uncertainty is substantially driven by the cloud parameters and aerosol correction...

...To conclude, the uncertainty in the AMF calculation is estimated to be in the 10-45% range for polluted conditions, leading to a total uncertainty in the tropospheric $NO_2$ columns likely in the range of $30-70\%$.

At its current form, the comparison with MAX-DOAS data does not tell much information regarding the causes of satellite errors (especially slope and bias), thus providing little new knowledge beyond previous findings.

Since the main goal of this work is the algorithm improvement rather than a validation study, we intended to present a new algorithm and show how it is an improvement wrt the previous version. We agree with the need of a better description, in particular in terms of the slope and bias, leading to the update of validation section below.

The discussion on sampling difference and retrieval algorithm should be aided with quantitative tests (of the algorithm assumptions/parameters, within the context of this study), particularly on how the data sampling (50 km is a relatively loose criterion; the temporal matching is not clear), cloud treatment and aerosol treatment each affects the comparison.

The choice of 50 km indeed seems loose at first glance, but considering the coarse resolution of GOME-2 pixels ($80\times40/40\times40$ km$^2$), it is not extreme. As explained above, the main goal of this section is not to do the best validation possible (this criterion has been tested in the past by e.g. Wang et al. (2017); Pinardi et al. (2015) and is included in a validation paper in preparation (Pinardi et al., 2018/2019)) but to compare the validation results for the 2 GDP versions for a fixed set of settings.

We have added the description of temporal matching in page 25 line 3:

The original ground-based MAXDOAS data usually retrieves $NO_2$ columns all day long every 20 to 30 minutes, and these values are linearly interpolated to the GOME-2 overpass time (9:30 local time), if original data exist within +/-1 hours.

The cloud treatment and aerosol treatment are the same between GDP 4.8 and 4.9 and thus leave almost no different impact during the comparison (Sect. 6.4 has been removed to avoid confusion, see response to the major comment 2).

Also, the main text only emphasizes the comparison at Xianghe (Figs. 14-16), which gives best consistency. However, the SI shows much poorer comparisons (in terms of bias and slope) at other sites. A better description in the main text of these comparisons (e.g., by better reference to SI, or moving some figures from SI to the main text) is needed to accurately present the overall quality of the satellite product.

We agree with the need to emphasize the results from other stations and the discussions in terms of bias and slope. Therefore, we have added more discussions of

the figures in the supplement and have rewritten page 26-28 with better comments at the other sites with a more quantitative explanation of bias and slope. We have updated page 26 line 1-page 27 line 3:

[revised manuscript text omitted]

Title: change "total" to "slant". The term "total column" is misleading.

As defined by EUMETSAT AC SAF product navigator (https://acsaf.org/produ cts/nto_no2.html), the name "total and tropospheric $NO_2$ column" is used in the previous retrieval algorithm descriptions (Valks et al., 2011, 2017) and therefore is preferable for this manuscript as a heritage algorithm.

P1, L18: Please give the full name of VLIDORT and clarify which version

Done.

P1, L20: "a large effect" is somehow ambiguous. Does it mean decrease or increase?

We have updated the sentence to:

A large effect (mainly enhancement in summer and reduction in winter) on the retrieved tropospheric $NO_2$ columns by more than 10% is found over polluted regions.

P2,L6 VOC is not necessary for aerosol formation from NOx.

We have updated to:

serve as a precursor of zone in the presence of volatile organic compounds (VOC) and of secondary aerosol through gas-to-particle conversion (Seinfeld et al., 1998).

The title and content of Sect. 3 should be clarified such that the reader understands the section is describing the old algorithm.

We have updated to:

Total and tropospheric $NO_2$ retrieval for GDP 4.8

P3,L16-17 the treatment of aerosol is crucial especially for polluted cases.

See response to major comment 1.

Table 1 check the formulas of intensity offset for QA4ECV

To make the comparison less confusing, we have removed the formulas from the table and classified the intensity offset in general as "constant" vs. "linear" in wavelength.

Figure 2, right panel discuss the discontinuity in 2015

Unfortunately we have not found any document that explicitly states a processor update, key-data change, or in-orbit operation during this period. One possible explanation would be the occurrence of a solar eclipse event on 20 March 2015, which might disturb the thermal environment of the instrument/platform.

Figure 6 the error appears to be stabilized after 2010. Please discuss.

We have added discussions in page 12 line 4:

...instrument degradation (Dikty et al., 2011; Munro et al., 2016) until the major throughput test in September 2009 (see Sect. 4.3.1) and stabilize afterwards.

P15,L10 "latitudinal" should be "longitudinal"

Done.

P19, the paragraph "Figure 11 illustrates..." what is the impact on cloud retrieval, and how would this affect the sampling (with criterion of CRF <0.5)?

See response to major comment 2 and 4.

P22, L8: All discussion in this section only shows the improvement of cloud retrieval itself. How about its influence on $NO_2$ retrieval?

See response to major comment 2.

P25,L3 test the choice of 50 km, which is relatively loose. Also, how is the temporal interpolation done?
P26, first paragraph need a better description of comparisons at other sites (see my major comment)
P26,L8-11 The explanation is qualitative. A more quantitative explanation of slope and bias is needed. For example, to what extent the slope and bias can be explained by sampling difference and algorithm limitations? See my major comment.

See response to major comment about validation.

P28,L9-10 the sentence is not clear.

We have removed page 28 line 7-10, as it will be discussed in the following paragraph.

P28,L17-21 again, the statement is too general.

See response to major comment about validation.

P29,L32 the uncertainty in tropospheric $NO_2$ AMF appears underestimated, especially given the large error in cloud and aerosol treatment (e.g., Lorente et al., 2017).

As described by Valks et al. (2011) for GDP 4.4, the estimated uncertainties in tropospheric $NO_2$ AMF is in the 15-50% range for polluted conditions with an average of ~33%. Considering the values 12-42% suggested by Lorente et al. (2017)

and all the improvements in the AMF calculation in GDP 4.9, we have updated the improved uncertainties in AMF to the 10-45% range and thus the total uncertainties in tropospheric $NO_2$ columns to the 30-70% range.

P30, first paragraph the consistency at other sites is much poorer. This point should be presented here.

We have updated page 30 line 4-7:

[revised manuscript text omitted]

---

## Author Response (AR2)

**Response to Referee #2**

Our response to the comments and detailed changes made to the manuscript version 2 are described below in black and blue, respectively, including the reviewer's text in red.

1. The reply on the use of surface albedo climatology should be reflected in the text.

We have extended the discussion for surface albedo in page 6 line 20:

Another advantage of the GOME-2 LER climatology is the use of more recent observations to reduce the errors introduced by ignoring the interannual variability of surface albedo, which are possibly large for varying snow and ice situations. Possible corrections for the surface albedo from a climatology include the use of external information about the actual snow and ice conditions, e.g., from Near-real-time Ice and Snow Extent (NISE) dataset (Nolin et al., 2005).

2. Define "total" column in the abstract and introduction to avoid confusion.

We have added in line 5 page 1:

, based on which initial total  $NO_2$  columns are computed using stratospheric air mass factors (AMFs).

We have added in line 3 page 3:

The total  $NO_2$  slant columns depend on the viewing geometry and also on parameters such as surface albedo and the presence of clouds and aerosol loads. They are therefore converted to initial total  $NO_2$  vertical columns through division by a stratospheric airmass factor.

3. Move the discussion on uncertainty to a new subsection (i.e., Sect. 6.5), so that the reader can easily locate this important information.

Done.

4. Many paragraphs are lengthy (e.g., many paragraphs in the sections for uncertainty analysis and comparison with MAX-DOAS data). Please split the paragraphs to enhance readability.

We have splitted the lengthy paragraphs in introduction, uncertainty analysis section, and validation section, as marked by  $\leftarrow$  symbol.

[revised manuscript text omitted]